# *Aspergillus fumigatus* mitogen-activated protein kinase MpkA is involved in gliotoxin production and self-protection

Patrícia Alves de Castro[1], Camila Figueiredo Pinzan[1], Thaila Fernanda dos Reis[1], Clara Valero[1,2], Norman Van Rhijn [2], Carla Menegatti[1], Ivan Lucas de Freitas Migliorini [1], Michael Bromley [2], Alastair B. Fleming[3], Aimee M. Traynor[4], Özlem Sarikaya-Bayram[4], Özgür Bayram [4] ✉, Iran Malavazi [5], Frank Ebel[6], Júlio César Jerônimo Barbosa [7], Taícia Fill[7], Monica Tallarico Pupo [1] & Gustavo H. Goldman [1] ✉

*Aspergillus fumigatus* is a saprophytic fungus that can cause a variety of human diseases known as aspergillosis. Mycotoxin gliotoxin (GT) production is important for its virulence and must be tightly regulated to avoid excess production and toxicity to the fungus. GT self-protection by GliT oxidoreductase and GtmA methyltransferase activities is related to the subcellular localization of these enzymes and how GT can be sequestered from the cytoplasm to avoid increased cell damage. Here, we show that GliT:GFP and GtmA:GFP are localized in the cytoplasm and in vacuoles during GT production. The Mitogen-Activated Protein kinase MpkA is essential for GT production and self-protection, interacts physically with GliT and GtmA and it is necessary for their regulation and subsequent presence in the vacuoles. The sensor histidine kinase SlnA^SLn1 is important for modulation of MpkA phosphorylation. Our work emphasizes the importance of MpkA and compartmentalization of cellular events for GT production and self-defense.

Fungal secondary metabolites (SMs) are important genetic traits for fungal survival and fitness in specific niche conditions. The genes that encode SMs are generally organized in biosynthetic gene clusters (BGCs)[1]. SMs are used in signaling, in mechanisms of antagonism, and to cause damage and facilitate the infection in animal and plant hosts, protecting the fungus from host immune cells, or mediating the acquisition of essential nutrients[2–9]. One of the best studied fungal SMs is gliotoxin (GT), a sulfur-containing mycotoxin, a member of the epipolythiopiperazines, produced by different fungal species, including *Gliocadium fimbriatum* (from which it was

originally isolated and named accordingly), *Aspergillus fumigatus* and closely related non-pathogenic species[10,11], and also by species of *Trichoderma* and *Penicillium*[12–15]. GT and other SMs can also cause toxicity to the producer fungus and mechanisms of self-protection and self-defense genes have evolved[16], such as: (i) the presence of more resistant target genes, like in the *A. fumigatus* fumagillin BGC, that has at least five copies of the methionine aminopeptidase, two of them present within the BGC and three in other regions of the genome; (i) presence of efflux transporters, such as the major facilitator superfamily transporter GliA, which belongs to the *A. fumigatus* GT

[1]Faculdade de Ciências Farmacêuticas de Ribeirão Preto, Universidade de São Paulo, Ribeirão Preto, Brazil. [2]Manchester Fungal Infection Group, Division of Evolution, Infection and Genomic Sciences, School of Biological Sciences, Faculty of Biology, Medicine and Health, University of Manchester, Manchester, United Kingdom. [3]Department of Microbiology, Moyne Institute of Preventive Medicine, Trinity College Dublin, Dublin, Ireland. [4]Department of Biology, Maynooth University, Maynooth, Co. Kildare, Ireland. [5]Departamento de Genética e Evolução, Centro de Ciências Biológicas e da Saúde, Universidade Federal de São Carlos, São Carlos, São Paulo, Brazil. [6]Institut für Infektionsmedizin und Zoonosen, Medizinische Fakultät, LMU, 80539 München, Germany. [7]Instituto de Química, Universidade Estadual de Campinas, Campinas, Brazil. ✉e-mail: Ozgur.Bayram@mu.ie; ggoldman@usp.br

BGC; and (iii) enzymes that modify the SMs, such as the reversible enzymatic activity of the oxidoreductase GliT that attenuates GT production by producing dithiol-GT (dtGT) with its subsequent conversion to bis(methylthio)gliotoxin (bmGT) by the *S*-adenosylmethionine-dependent gliotoxin *bis*-thiomethyltransferase GtmA[16]. Removal of these GT self-defense genes by knock-out causes increased susceptibility to GT[15,17–22].

*A. fumigatus* is a saprophytic fungus that can cause a variety of human and animal diseases known as aspergillosis[23], producing GT as an important genetic determinant of virulence[24,25]. GT has already been detected in vivo in murine models of invasive aspergillosis (IA), in human cancer patients[24], and produced by isolates derived from patients with COVID-19 associated pulmonary aspergillosis[25]. The GT cluster is located on chromosome VI in *A. fumigatus* Af293 and contains 13 *gli* genes responsible for GT biosynthesis and secretion[15]. The biosynthesis of GT is exquisitely regulated since there is crosstalk between different cellular pathways, such as sulfur metabolism [cysteine (Cys) and methionine (Met)], oxidative stress defenses [glutathione (GSH) and ergothioneine (EGT)], methylation [S-adenosylmethionine (SAM)], and iron metabolism (Fe–S clusters)[26–30]. Several signal transduction pathways involving transcription factors (TFs), protein kinases, regulators of G-protein signaling as well as chromatin modifying enzymes are involved in the regulation of GT biosynthesis[15]. In the mammalian host, GT has an immunosuppresive role by: (i) interfering with macrophage-mediated phagocytosis through prevention of integrin activation and actin dynamics, resulting in macrophage membrane retraction and failure to phagocytose pathogen targets[31]; (ii) inhibiting the production of pro-inflammatory cytokines secreted by macrophages and the activation of the NFkappaB regulatory complex[32]; (iii) disrupting the correct assembly of NADPH oxidase through preventing p47phox phosphorylation and cytoskeletal incorporation as well as membrane translocation of subunits p47phox, p67phox and p40phox[33]; and (iv) suppressing neutrophil chemoattraction by targeting the activity of leukotriene A4 (LTA4) hydrolase, an enzyme that participates in LTA biosynthesis[34]. *A. fumigatus* GT production and regulation must be tightly regulated to avoid excess production and toxicity to the fungus. GliT produces the final toxic form of GT by catalyzing disulfide bridge closure of the precursor dithiol gliotoxin (dtGT)[15,20]. However, if there is excess GT production, GliT can reduce GSH and produce dtGT, attenuating GT toxicity[15,20]. GtmA, whose gene is not located in the GT BGC, is able to convert dtGT into bisdethiobis(methylthio)-gliotoxin (bmGT) and to attenuate GT production postbiosynthetically[17,18,21,22]. It is thought that the primary role of GtmA is a decrease in GT biosynthesis and not a backup for GliT and toxin neutralization[15]. Recently, we have identified the TF RglT as the main regulator of GliT and GtmA through directly binding to their promoter regions during GT-producing and self-protection conditions[35]. We also screened an *A. fumigatus* deletion library of 484 TFs for GT sensitivity and identified 15 TFs important for GT self-protection[36]. Of these, KojR, which is essential for kojic acid biosynthesis in *Aspergillus oryzae*, was also crucial for virulence and GT biosynthesis in *A. fumigatus*, and for GT protection in *A. fumigatus*.

An important aspect of the *A. fumigatus* GT production and self-protection by GliT and GtmA is related to the cellular sub-location of these enzymes and how GT can be temporarily sequestered from the cytoplasm to avoid increased cell damage. It has been speculated that reduced dithiol gliotoxin (dtGT) may be sequestered into intracellular vesicles where it is converted to the oxidized form, by an unidentified activity, prior to release from the cell by an exocytotic mechanism complementary to GliA-mediated efflux[20]. Therefore, the aim of this work is to elucidate GliT and GtmA subcellular localization under GT production and self-protection. We demonstrate that both proteins are localized in the cytoplasm and enriched in vacuolar structures during GT production while are present in structures that resemble the intracellular endomembrane network and/or endocytosis/exocytosis

vesicles. To gain more information about metabolic pathways involved in the GliT and GtmA regulation, we immunoprecipitated these two proteins under GT producing conditions and identified several regulatory proteins as interacting with GliT and GtmA, among them the mitogen-activated protein kinase MpkA. We demonstrated that Δ*mpkA* is not able to produce either GT or bmGT and is highly sensitive to GT. MpkA controls positively most of the *gli* genes mRNA accumulation during either GT production or self-protection. MpkA is required for GliT and GtmA protein levels and for their localization into the vacuoles during GT production. Not only is MpkA involved in GT self-protection but also other protein kinases are involved in this process since a screening of a collection of 110 non-essential protein kinase deletion mutants revealed six and one mutants as more GT-susceptible and –resistant, respectively. One of these susceptible mutants corresponds to the sensor histidine kinase SlnA[Sln1] and we demonstrate that SlnA is important for GT production but it is influencing MpkA phosphorylation to the same extent than the wild-type, suggesting SlnA is controlling MpkA-independent pathway(s) involved in GT production. However, SlnA is important for the modulation of the MpkA phosphorylation during GT self-protection, suggesting SlnA is one of the possible receptors involved in the MpkA regulation of GT self-protection. In addition to GliT and GtmA regulation and sub-cellular localization, we also investigated the involvement of peroxisomes on GT self-protection, we characterized the PexE[Pex5] and PexG[Pex7] peroxisome receptors, and how they mediate GT self-protection. We demonstrate that both receptors are involved in the regulation of GT production but only *A. fumigatus* Δ*pexE* mutant is GT-sensitive and has attenuated virulence in a murine model of invasive pulmonary aspergillosis (IPA).

## Results

### GliT:GFP and GtmA:GFP are localized in vacuoles, vesicles, endoplasmic reticulum, and in the cytoplasm during GT production and self-defense

As an initial step to characterize the GliT and GtmA regulation, we characterize their localization under producing (24 to 72 h growth in Czapek-Dox liquid medium) and self-protection (germlings previously grown in GT non-producing liquid medium were exposed to GT 3 μg/mL) conditions. Protein localization with longer time points, such as 48 and 72 h was technically challenging due to hyphal overgrowth, therefore a timepoint of 24 h was selected to conduct the microscopy experiments. After 24 h in GT-production conditions, the wild-type has not shown any auto-fluorescence in the GFP excitation and emission spectra (Fig. 1a, b) while the GliT:GFP and GtmA:GFP were localized in the cytoplasm (100 % of the germlings) and had enriched localization in 89.77 and 97.25 % of the germlings, respectively (Fig. 1a) of structures that resemble to vacuoles (Fig. 1b, c). This was confirmed by co-staining with CellTracker Blue CMAC Dye (7-amino-4-chloromethylcoumarin; https://www.thermofisher.com/order/catalog/product/C2110) which accumulates in the vacuoles (Fig. 1b, c). GliT:GFP and GtmA:GFP germlings were grown for 24 h in MM and exposed to GT 3 μg/ml for 2 h and co-stained with either CMAC, ER-Tracker, or FM4-64 Dye [*N*-(3-Triethylammoniumpropyl)−4-(6-(4-(Diethylamino) Phenyl) Hexatrienyl) Pyridinium Dibromide] (Fig. 2). ER-Tracker Blue-White DPX dye is a photostable probe that is selective for the endoplasmic reticulum (ER) in live cells (https://www.thermofisher.com/order/catalog/product/E12353). FM4-64 is recommended for staining vacuolar membranes and for studying the endocytic pathway;[37,38] https://www.thermofisher.com/order/catalog/product/T13320). Upon GT exposure, GliT:GFP accumulates in the cytoplasm, but not in the GT-free control, without any co-localization with CMAC (Fig. 2a, b) but with 100% germlings with ER-tracker (Fig. 2a, c) and 100 % germlings with FM4-64 (Fig. 2a, d). GtmA:GFP showed no fluorescence signal in the control but when the germlings were exposed to GT, there was diffuse and 100% germlings with punctuated distribution in the cytoplasm (Fig. 2a, e–g). These punctuated structures were not observed in the

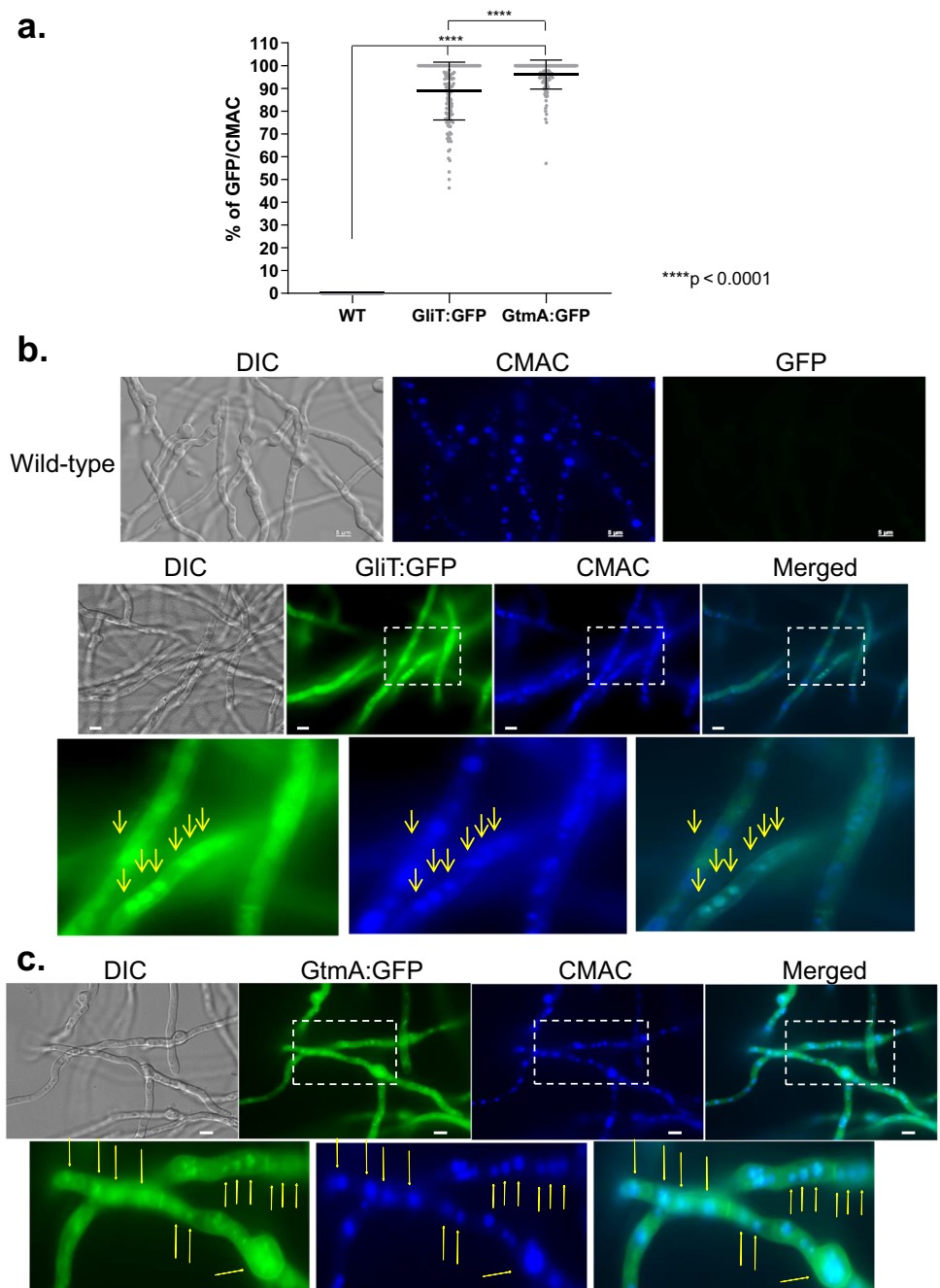

**Fig. 1 | GliT:GFP and GtmA:GFP have enriched vacuolar localization during GT production.** GliT:GFP and GtmA:GFP germlings were grown in liquid Czapek-dox medium for 24 h at 37 °C. Representative brightfield, differential interference contrast (DIC), and Cell tracker Blue CMAC (CellTracker Blue CMAC Dye (7-amino-4-chloromethylcoumarin) for vacuolar staining. **a** The number of GliT:GFP and GtmA:GFP germlings that co-localized with CMAC that was used for vacuolar staining were determined. We have counted three independent experiments with 45 germlings for each strain per experiment ($N = 135$ germlings) and the results were expressed as the mean values (%) of 3 independent experiments of GFP that co-localizes with CMAC. Yellow arrows indicate the localization of GliT:GFP and GtmA:GFP in magnified images of sub-cellular structures. The $p$-values were calculated using One-way ANOVA with Tukey's multiple comparisons test, ****$p < 0.0001$. Germlings were assigned as positively present in the vacuoles if they have concomitant localization of GliT:GFP or GtmA:GFP in the same vacuole stained by CMC in the same germling. Representative images are shown in (**b**) and (**c**). Bars, 5 μm. Source data are provided as a Source Data file.

GliT:GFP (Fig. 2a). Upon GT exposure, GtmA:GFP did not co-localize with CMAC (Figs. 2a and 2e), but 10 % germlings co-localized with ER-tracker (Fig. 2a, f) and 100 % germlings with FM4-64 (Fig. 2a, g).

Taken together these results suggest that GliT:GFP and GtmA:GFP have different protein localizations during the early phase of GT production and GT self-defense. Upon GT-production conditions both proteins accumulated either in the cytoplasm or in structures that resembled to vacuoles. In contrast, in GT-defense conditions GliT and GtmA showed cytoplasmatic accumulation and some aggregation in structures that could be related to the intracellular endomembrane network, endoplasmic reticulum, and/or endocytosis/exocytosis vesicles.

## GliT and GtmA protein interactions during GT production
As a preliminary step to understand metabolic pathways involved in the GliT and GtmA regulation, GliT:GFP and GtmA:GFP were

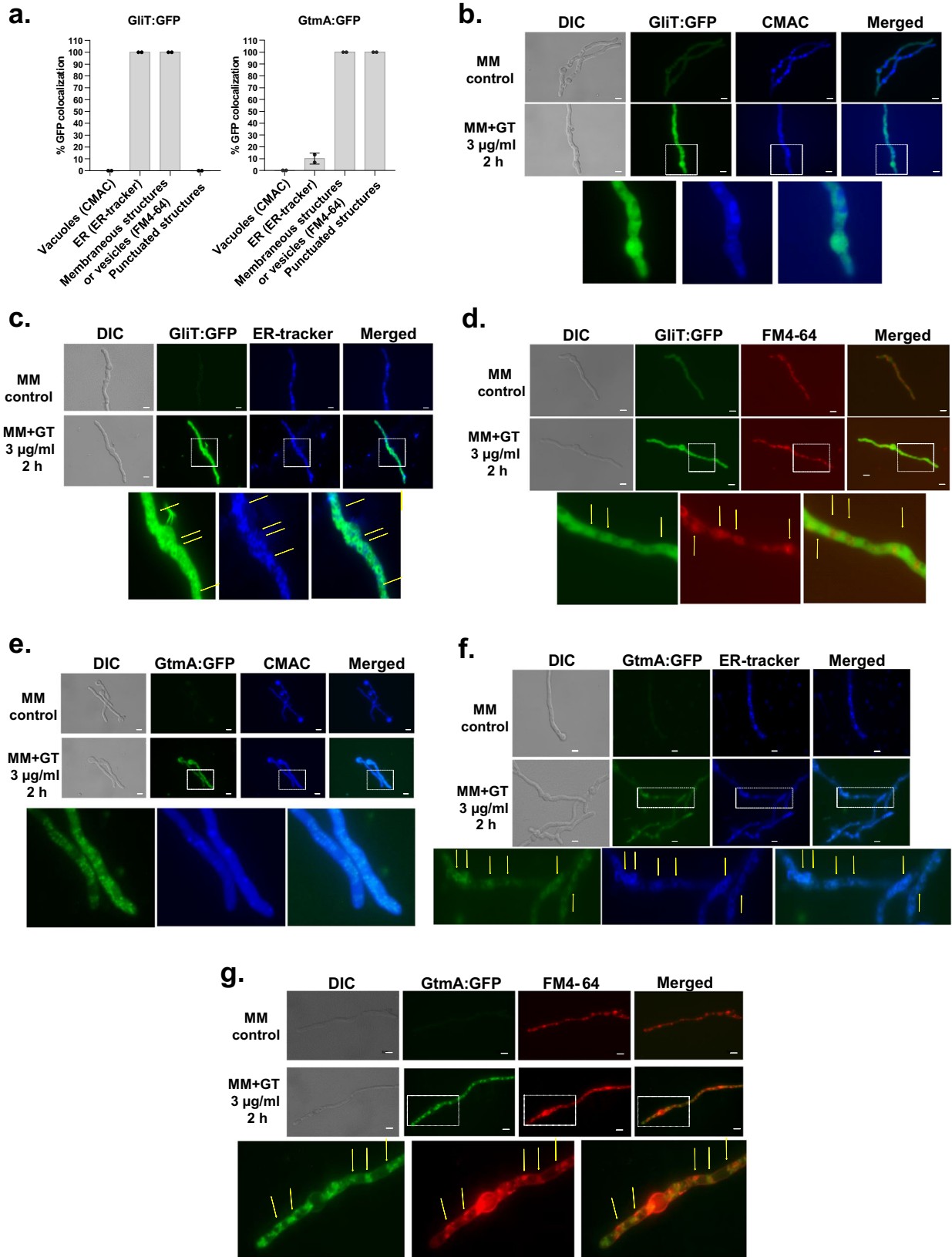

immunoprecipitated (IP) during 24, 48, and 72 h in GT production conditions. During 24 to 72 h GT production, after subtraction of the proteins non-specifically immunoprecipitated of the wild-type strain, 133 proteins that associate with GliT:GFP and 203 proteins with GtmA:GFP were identified (Fig. 3a). Seventy-one proteins were found to be in association with both GliT:GFP and GtmA:GFP while 62 and 132

were unique for GliT:GFP and GtmA:GFP, respectively (Fig. 3a). Functional categorization (FunCat) (https://elbe.hki-jena.de/fungifun/fungifun.php) analyzes (p-value < 0.05) of all significantly IP proteins showed enrichment for (i) GliT:GFP in purine nucleotide/nucleoside anabolism, pyridoxal phosphate binding, amino acid metabolism, and stress response; (ii) GtmA:GFP in receptor mediated signaling, purine

**Fig. 2 | GliT:GFP and GtmA:GFP localize in the cytoplasm, endoplasmic reticulum, and small vesicle structures during GT self-protection.** GliT:GFP and GtmA:GFP germlings were grown in liquid MM for 24 h at 37 °C and exposed to GT 3 μg/ml for 2 h and co-stained with either CMAC, ER-Tracker Blue-White DPX, or FM4-64 **a** The number of GliT:GFP and GtmA:GFP germlings that co-localized with CMAC, ER-tracker, or FM4-64 staining were determined. We have counted two independent experiments with 45 germlings per experiment (N = 90 germlings) for each strain and the results were expressed as mean values (%) of the 2 independent experiments of GFP germlings that co-localizes with vacuoles (CMAC), Endoplasmic Reticulum, ER (ER-tracker) or vesicles (FM4-64). The percentage of germlings with punctuated structures of GliT:GFP and GtmA:GFP were also determined. Yellow arrows indicate the localization of GliT:GFP and GtmA:GFP in magnified images of sub-cellular structures. Germlings were assigned as positively present in the vacuoles, endoplasmic reticulum, vesicles, or endomembranes if at least one of these structures was co-localizing with GliT:GFP or GtmA:GFP in the same germling. Representative images are shown in (**b**), (**c**), (**d**), (**e**), (**f**), and (**g**). Bars, 5 μm. Source data are provided as a Source Data file.

nucleotide/nucleoside anabolism, NAD/NADP binding, amino acid metabolism, and stress response; and (iii) both GliT:GFP and GtmA:GFP in pyridoxal phosphate binding, purine/nucleoside binding, amino acid metabolism, and stress response (Fig. 3b–d and Supplementary Data S1).

Among all the GliT:GFP and GtmA:GFP IP proteins, of particular interest were the other enzymes in the *gli* pathway, such as GliN and GliG (in either GliT and/or GtmA), and GtmA IP by GliT but not GliT IP by GtmA (Table 1 and Supplementary Data S1). Several protein kinases and phosphatases were also pulled down including, Mitogen-Activated protein kinases (MAPK), MpkA, MpkB, and SakA, two calcium/calmodulin-dependent protein kinases, and GskA (either IP by GliT:GFP and/or GtmA:GFP), and protein phosphatases, such as PphB and PtcB (IP by GtmA:GFP). Considering the impact of oxidative stress and glutathione metabolism in the conversion from GT to dtGT (and subsequently to bmGT), it is worth noting the IP of glutathione peroxidase, lactoylglutathione lyase, cystathionine beta-lyase (that plays a role in de novo L-methionine biosynthesis process), peptide-methionine(S)-S-oxireductase, and FhpA, a flavohemoprotein (Table 1).

Taken together these results suggest that GliT and GtmA are interacting with several protein kinases and phosphatases and enzymes involved in the glutathione metabolism and oxidative stress response during GT-production.

## MAP kinase MpkA is important for GT production and self-defense

Considering that *A. fumigatus* MpkA has previously been described as being involved in GT regulation and production[39], we decided to validate and investigate the role played by this MAPK in GliT and GtmA regulation. To further support our identification of GliT:GFP and GtmA:GFP as interactors with MpkA, we carried out a co-immunoprecipitation analysis using epitope-tagged forms of these two proteins. A functional 3xHA-tagged form of MpkA was introduced into these strains (Supplementary Fig. S1). Isogenic GliT:GFP and GtmA:GFP strains either containing or lacking the MpkA:3xHA allele were grown under GT production conditions or grown overnight and exposed to 5 μg/mL of GT for 30 min, 1 h, and 3 h, native protein extracts were prepared, and the MpkA:3xHA protein was recovered by IP with anti-mouse HA antibody. These anti-HA immunoprecipitations were electrophoresed on SDS-PAGE and then analyzed by Western blotting using anti-HA antibodies (Fig. 3e). Only when both the MpkA:3xHA and the GliT:GFP and GtmA:GFP fusions were present was co-immunoprecipitation seen. When GliT:GFP was IP by using anti-GFP and revealed by anti-HA, under GT production conditions, there is surprisingly only seen in 24 h production and under GT self-protection this is seen in all four time points including the control (Fig. 3e). When GtmA:GFP was IP by using anti-GFP and revealed by anti-HA, under GT production conditions there are bands in all three times points. (Fig. 3f). Under GT self-protection, MpkA:3xHA is seen in all four time points including the control (Fig. 3f). Expression of only the GliT:GFP or GtmA:GFP fusion proteins did not show any evidence for nonspecific recovery of this factor by the anti-HA antibody. By using Net-Phos-3.1 (https://services.healthtech.dtu.dk/service.php?NetPhos-3.1), we were able to predict three putative threonine p38MAPK phosphorylation sites at residues 278, 297, and 303 of GliT but no significant phosphorylated residues at GtmA. The functionality of these putative phosphorylation sites remains to be investigated. These data strongly support the view that GliT and MpkA associate in vivo during GT production and GT self-protection while GtmA associates only during GT self-protection.

The Δ*mpkA* has a strong growth defect and its complementation by *mpkA*+ restores this phenotype to the wild-type growth (Fig. 4a). Although the Δ*mpkA* mutant has a strong growth defect in solid medium with significantly reduced radial growth, surprisingly its growth in liquid Czapek-dox medium after 72 h is not statistically different from the wild-type (wild-type, 0.321 ± 0.014 mg and Δ*mpkA*, 0.303 ± 0.005 mg). After 72 h growth in Czapek-dox medium the Δ*mpkA* mutant does not produce either GT or bmGT (Fig. 4b). Since other MAP kinases (like SakA and MpkC, Table 1) were shown to be interacting either with GliT and/or GtmA, we also investigated the GT production in the corresponding null mutants, including another MAPK mutant, Δ*mpkB*, and the double mutant Δ*sakA* Δ*mpkC*. All the mutants showed comparable GT and bmGT production with the wild-type, except for the Δ*sakA* Δ*mpkC* mutant that has reduced bmGT production (Fig. 4b). The Δ*mpkA* mutant has reduced growth in liquid medium in the presence of GT (70 μg/mL) when compared to the wild-type and complemented strains as quantified by Alamar blue (Fig. 4c). The MpkA is more phosphorylated during GT production and self-protection (Fig. 4d). The GliT:GFP and GtmA:GFP protein are highly produced from 24 to 72 h while both proteins have increased production when the strains are exposed to GT (5 μg/mL) for 3 h; however, the GliT production is much higher than GtmA:GFP (Fig. 4d). The Δ*mpkA* has reduced mRNA accumulation of most of the *gli* genes during GT production and self-protection (except for *gliF*, *gliT*, *gliJ*, and *gtmA*) (Fig. 4e, f). We have not observed any differences in vacuolar formation in the Δ*mpkA* when compared to the wild-type strain (Supplementary Fig. S2). We hypothesized the MpkA could affect the GliT:GFP and GtmA:GFP protein levels and subsequent translocation to the vacuoles. Functional Δ*mpkA* GliT:GFP and Δ*mpkA* GtmA:GFP strains showed reduced GliT:GFP and GtmA:GFP levels under GT production and vacuolar translocation frequencies (Fig. 4d and Supplementary Fig. S2).

Recently, we identified two transcription factors, RglT and KojR, that were demonstrated to be essential for the regulation of GliT and GtmA, and GT production[35,36]. The Δ*mpkA* has reduced and increased mRNA accumulation of *rglT* during GT production and self-protection, respectively, while *kojR* has decreased mRNA accumulation at 72 h GT production but no differences to the wild-type during GT self-protection (Fig. 4g).

Taken together these results indicate that MpkA is essential for GT production and self-protection, positively regulating *gliT* and *gtmA* mRNA and protein accumulation during GT production and negatively regulating during GT self-protection. Moreover, MpkA positively and negatively regulates *rglT* mRNA accumulation during GT-production and self-protection, respectively.

## Screening of protein kinases involved in gliotoxin self-defense

Aiming to identify additional protein kinases (PK) involved in gliotoxin self-protection, we performed a screening looking for

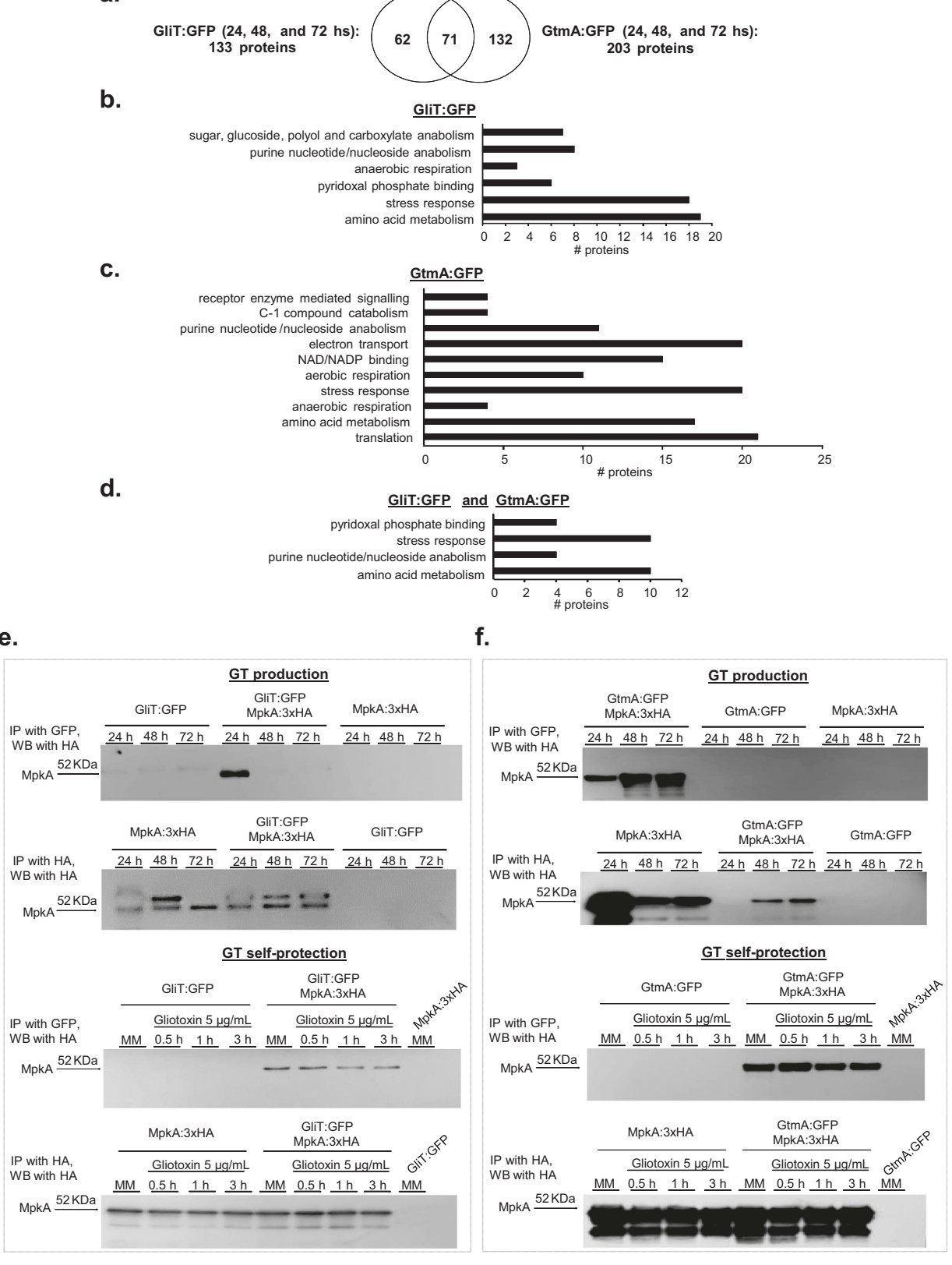

**Fig. 3 | Identification of proteins that interact with GliT:GFP and GtmA:GFP during GT production.** *A. fumigatus* was grown for 24, 48, and 72 h at 37 °C in liquid Czapek-dox medium. Proteins were extracted and immunoprecipitated (IP) with anti-GFP antibody. **a** Venn diagram showing the number of unique and shared IP proteins identified by mass spectrometry as interacting with GliT:GFP and/or GtmA:GFP. **b**–**d** Fungifun categorization of proteins interacting with GliT:GFP, GtmA:GFP, and GliT:GFP and GtmA:GFP. (**e** and **f**) Co-IPs for GliT:GFP, GtmA:GFP,

GliT:GFP MpkA:3xHA, GtmA:GFP MpkA:3xHA and MpkA:3xHA strains under GT production and self-protection. For GT production, *A. fumigatus* was grown under the same conditions described in (**a**) while for GT self-protection *A. fumigatus* was grown in liquid minimal medium for 24 h at 37 °C and exposed or not to GT 5 µg/mL for 30 min, 1 h and 3 h. The western blots are representative results from 2 independent experiments. Source data are provided as a Source Data file.

**Table 1 | Proteins that are possibly associated with *A. fumigatus* GliT:GFP and GtmA:GFP under GT production**

| Accession | Description | |
|---|---|---|
| | Gliotoxin pathway | |
| Afu6g09740 | GliT, gliotoxin sulfhydryl oxidase required for gliotoxin biosynthesis | 24, 48, and 72 h, GliT and 48 and 72 h, GtmA |
| Afu6g09720 | GliN, methyltransferase, encoded in the putative gliotoxin biosynthetic gene cluster | 48 h, GliT and 48, 72 h GtmA |
| Afu6g09690 | GliG, glutathione S-transferase encoded in the gliotoxin biosynthetic gene cluster | 48 h, GtmA |
| Afu2g11120 | GtmA, methyltransferase activity | 24, 48, and 72 h GtmA |
| | Protein kinases | |
| Afu4g13720 | MpkA, mitogen-activated protein kinase (MAPK) | 48 and 72 h, GliT and 48 h, GtmA |
| Afu6g12820 | MpkB, mitogen-activated protein kinase (MAPK) | 72 h, GliT |
| Afu1g12940 | SakA, mitogen-activated protein kinase (MAPK) | 24 h, GliT and 24 h and 48 h, GtmA |
| Afu2g13680 | Calcium/calmodulin-dependent protein kinase | 48 h, GtmA |
| Afu2g03490 | Calcium/calmodulin-dependent protein kinase | 48 h, GliT and 48 h, GtmA |
| Afu6g05120 | GskA, protein serine/threonine kinase activity | 48 h, GliT and 48, 72 h GtmA |
| | Protein phosphatases | |
| Afu6g10830 | PphB, protein serine/threonine phosphatase activity | 48 h, GtmA |
| Afu1g09280 | PtcB, type 2 C protein phosphatase (PP2C) involved in dephosphorylation of SakA MAP kinase | 24 and 48 h, GtmA |
| | Heat-shock proteins | |
| Afu3g14540 | hsp30, 30-kilodalton heat shock protein | 24 h, GliT and 48 h GtmA |
| Afu5g13920 | Wos2, Hsp90 binding co-chaperone | 72 h, GliT |
| | Glutathione metabolismo | |
| Afu3g12270 | Glutathione peroxidase; peroxiredoxin | 72 h, GliT |
| Afu6g07940 | Lactoylglutathione lyase | 24 h, GliT |
| | Miscellaneous | |
| Afu2g02780 | Ortholog(s) have mRNA binding activity, role in negative regulation of MAPK cascade and cytosol localization | 48 h, GliT |
| Afu2g03140 | Peptide-methionine (S)-S-oxide reductase activity | 48 h, GtmA |
| Afu3g09320 | Serine hydroxymethyltransferase | 48 h, GliT and 48 h, GtmA |
| Afu4g03410 | FhpA, flavohemoprotein; protein induced by heat shock and hipoxia | 48 h, GliT |
| Afu4g03950 | Cystathionine beta-lyase activity, role in 'de novo' L-methionine biosynthetic process | 48 h, GliT and 48 h, GtmA |
| Afu4g11720 | Phosphatidyl synthase; protein levels increase in response to pkaC overexpression | 72 h, GliT |
| Afu6g08360 | Thiazole biosynthesis enzyme; hypoxia induced protein; induced by gliotoxin exposure | 72 h, GliT and 72 h GtmA |

differences in growth in the presence of MM + GT compared to MM by quantifying radial growth susceptibility. For this screening, 109 PK mutants[40] (Supplementary Table S1) were grown on solid MM and MM + 30 μg/mL of GT. In the primary screening we identified 11 PK mutants (5 and 6 with less growth and more growth, respectively) (Fig. 5a). Further validation screening was performed by growing these selected PK mutants in liquid MM in the absence and presence of GT 30 μg/mL and the metabolic activity was quantified by using Alamar blue (Fig. 5b). Upon Alamar blue validation, we identified four PK mutants with decreased radial growth in the presence of GT (Fig. 5b): AFUB_010510 (Δ*kin1*, regulates polarized exocytosis and the Ire1p-mediated unfolded protein response), AFUB_017740 (Δ*slnA*, transmembrane histidine phosphotransfer kinase and osmosensor; regulates MAP kinase cascade), AFUB_059390 (Δ*hrk1*, implicated in activation of the plasma membrane H(+)-ATPase Pma1p in response to glucose metabolism), and AFUB_070630 (Δ*mpkA*). Six mutants (AFUB_077790, AFUB_081540, AFUB_030570, AFUB_011380, AFUB_044400, and AFUB_074550) had increased growth in the presence of GT in both screenings but were not validated in the Alamar blue assay and were not considered here for further analysis.

We decided to further characterize *slnA* mutant because we have previously demonstrated that several kinds of stress, such as osmotic, cell wall, and oxidative stresses could impact Δ*slnA* growth[41]. The Δ*slnA* mutant has less GT and similar bmGT production than the wild-type strain, respectively (Fig. 5c). MpkA phosphorylation in the Δ*slnA* mutant has comparable levels to the wild-type during GT production

(Fig. 5d) but increased levels during GT protection (about 3-fold at 1 h exposure to 5 μg/mL GT, Fig. 5e). The Δ*slnA* mutant has higher *gliT* and *gtmA* mRNA accumulation upon GT production than the wild-type (Fig. 5f). Upon GT self-protection, *gliT* mRNA accumulation is lower and higher at 0.5 and 1 h exposure to 5 μg/mL GT than the wild-type strain, respectively (Fig. 5f); *gtmA* mRNA accumulation is also lower in the Δ*slnA* mutant upon GT self-protection conditions than the wild-type (Fig. 5f).

Taken together, these results suggest the SlnA is important for the modulation of MpkA phosphorylation, affecting the GT production levels and GT protection.

### *Aspergillus spp* peroxisome receptors PexE^Pex5 and PexG^Pex7 are involved in GT self-protection

Peroxisomes are responsible for detoxification, catabolism of linear and branched-chain fatty acid, and removal of $H_2O_2$ by catalases[42–53]. Many SMs are initially synthesized in the peroxisomes, such as aflatoxin and sterigmatocystin, as biosynthetic steps of many other SM pathways, such as paxilline, AK-toxin, penicillin, cephalosporin and some siderophores[16,44]. The accumulation of GT and dtGT has a great impact on the generation of reactive oxygen species by the cycling between the oxidized (disulfide) and reduced (dithiol) forms of the molecule[3]. We speculate that due to the important role played by peroxisomes in the oxiredox equilibrium of the cell, they could also be involved in *A. fumigatus* GT production and self-protection. Peroxisomal proteins, such as the peroxin receptors Pex5 and Pex7, have the

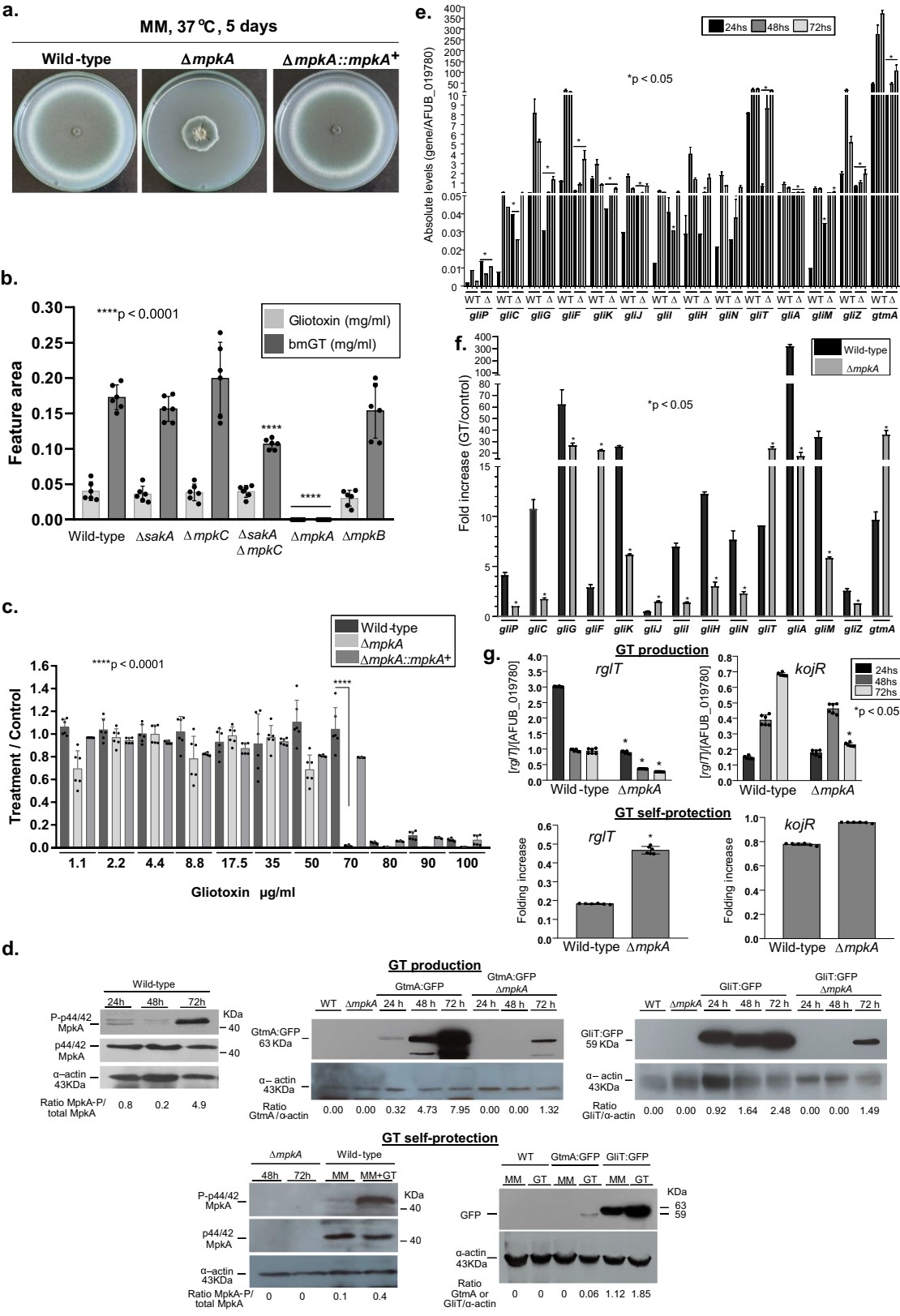

function to recruit, transport, and introduce the peroxisomal matrix proteins into the peroxisomes[42,43,54,55]. The matrix proteins contain the peroxisomal targeting signals PTS1 and/or PTS2 that are recognized by the peroxins Pex5 and Pex7, respectively. PTS1 is a tri-peptide with the consensus sequence (S/A/C)(K/H/R)(L/M) located at the extreme C-terminus, whereas PTS2 is a nona-peptide of the consensus sequence (R/K)(L/V/I)X5(H/Q)(L/A) located near the N-terminus of a matrix protein[42,43,54,55]. Although the GliT:GFP and GtmA:GFP subcellular localization at the vacuoles during GT production is unequivocal, there is a diffuse subcellular location to these proteins upon GT-self protection. Since there are some punctuated distribution of GtmA:GFP (Fig. 2), we raised the hypothesis that these proteins could be localized

**Fig. 4 | The mitogen-activated protein kinase (MAPK) MpkA is essential for GT production and self-protection. a** The wild-type, Δ*mpkA* and Δ*mpkA::mpkA*+ strains were grown for 5 days at 37 °C in solid minimal medium. **b** The wild-type and the MAPK mutants Δ*sakA*, Δ*mpkC*, Δ*sakA* Δ*mpkC*, Δ*mpkA*, and Δ*mpkB* were grown for 72 h at 37 °C in liquid Czapek-dox medium, secondary metabolites of the supernatant extracted and GT and bmGT identified. All the results are the mean of *n* = 6 biological replicates from 3 independent experiments ± SD, and *p*-values were calculated using One-way ANOVA with Tukey's multiple comparisons test. ****\**p* < 0.0001. **c** Metabolic activity expressed by Alamar blue of *A. fumigatus* grown for 48 h in the absence or presence of different concentrations of GT. All the results are the mean of n = 6 biological replicates from 3 independent experiments ± SD; (****, *p* < 0.0001; One-way ANOVA with Tukey's multiple comparisons test

comparing the Δ*mpkA*, and Δ*mpkA::mpkA*+ mutants with the wild-type strain). **d** Western blot analysis for GT production and self-protection. GliT:GFP and GtmA:GFP were identified by anti-GFP antibody, MpkA-P and total MpkA were identified by using P-p44/42 and p44/42 antibodies while actin by anti-actin antibody. For GT production *A. fumigatus* was grown under the same conditions described in (**b**) while for GT self-protection *A. fumigatus* was grow in liquid minimal medium for 24 h at 37 °C and exposed or not to GT 5 μg/mL for 3 h. **e, f** RTqPCR for genes of the GT pathway and *gtmA*. and (**g**) RTqPCR for *rglT* and *kojR*, under GT production in liquid Czapek-dox medium for 24, 28 and 72 h, and GT self-protection conditions as described in (**d**). Results are the mean of 3 biologically independent experiments (with two technical replicates) ± SD; (*, *p* < 0.05, *t*-test). Source data are provided as a Source Data file.

in the peroxisomes upon GT-self defense. Initially, we used several lysosome targeting signal predictors [such as psortII (https://psort. hgc.jp)], PTS1 prediction IMP bioinformatics group (https://mendel. imp.ac.at/pts1/) and DeepLoc-1.0 (https://services.healthtech.dtu.dk/ service.php?DeepLoc-1.0), aiming to predict if GliT and GtmA could be transported to the lysosomes. The results were ambiguous since only DeepLoc-10 predicts both proteins as being lysosomal (GliT=0.6683 and GtmA=0.4158).

We reasoned that the molecular characterization of *A. fumigatus* Pex5 and Pex7 homologs (here called PexE and PexG, respectively) could help us to assign a possible function for peroxisomes in GT production and detoxification. We deleted both homologs in *A. fumigatus* and extended this analysis for the GT non-producer *A. nidulans* that has a homolog for GliT but no homologs for GtmA[36]. We have previously demonstrated that GliT in both species is under the control of the transcription factor RglT[35]. *A. fumigatus* Δ*pexE* mutant showed reduced growth and conidiation, and it is very sensitive to GT (Fig. 6a, b). In contrast, *A. fumigatus* Δ*pexG* has a comparable phenotype to the wild-type strain (Fig. 6c). Both *A. nidulans* Δ*pexE* and Δ*pexG* are more sensitive to GT than the wild-type but similarly to *A. fumigatus*, only Δ*pexE* has a reduced growth and conidiation phenotypes (Fig. 6d–f). As previously shown[45], *A. nidulans* Δ*pexE* mutant have a partial biotin deficiency phenotype (Fig. 6d), but this partial auxotrophy is not observed in *A. fumigatus* Δ*pexE* mutant (Fig. 6a). Notice that we are using different GT concentrations for microscopy (3 μg/ml at Fig. 1) and A. *fumigatus* and *A. nidulans* radial growth experiments (30 μg/ml versus 5 μg/ml, Fig. 6) because germlings are much more sensitive to GT than the mycelia and *A. nidulans* is more GT-sensitive than *A. fumigatus*.

*A. fumigatus* Δ*pexE* mutants have a reduced GT (about 7.5- to 15-fold reduction; Fig. 6g, left graph) and bmGT production (about 15- to 3-fold reduction; Fig. 6g, right graph) while surprisingly Δ*pexG* mutants show a dramatic increase in GT production (about 25- to 50-fold compared to the wild-strain, Fig. 6h, left graph) and a bmGT production comparable with wild-type (Fig. 6h, right panel). Not only the SMs production is altered in the *pex* mutants but also several other SMs, such as brevianamide F, fumigaclavine C, pseurotin A, and fumagillin have increased or decreased production in these mutants when compared to the wild-type strain (Table 2). The *A. fumigatus* Δ*pexG* mutants are more sensitive to oxidative stress caused by menadione but not to *t*-butyl hydroperoxide, allyl alcohol, and hydrogen peroxide (Supplementary Fig. S3), and have no growth deficiencies when grown on cysteine or methionine as single sulfur sources (Supplementary Figure S3). No growth defects in the Δ*pexE* mutants exposed to the same growth conditions were observed (Supplementary Fig. S3). We also investigated if Δ*pexE* and Δ*pexG* could impact *A. fumigatus* virulence in a chemotherapeutic murine model of IPA. All mice infected with the wild-type and Δ*pexG* strains died between day 6 and 15 post-infection (p.i.), whereas 40 to 70 % of the mice infected with the Δ*pexE* survived for the duration of the experiment (Fig. 6i, j).

GT production is normally induced in Czapek-dox medium and *Aspergillus* minimal medium is regarded as a non-inducing

condition[36]. Due to the high levels of GT production by Δ*pexG* mutant, we comparatively investigated the mRNA levels of the *gli* genes and *gtmA*, and the GT and bmGT production under inducing (grown in Czapek-dox medium) and non-inducing (grown in *Aspergillus* minimal medium) conditions (Fig. 6 and Supplementary Fig. S4). mRNA levels were expressed as the fold increase of each gene from the Δ*pexG* mutant divided by the wild-type: fold increase ratios equal to 1, below 1 or above 1 corresponded to similar, lower or higher mRNA levels in the Δ*pexG* mutant than in the wild-type (Supplementary Fig. S4). In non-inducing conditions, all *gli* genes had higher than 1 expression ratios at 48 h production except for *gtmA* that had higher than 1 expression ratio at 72 h (Supplementary Fig. S4a). In inducing conditions, all *gli* genes and *gtmA* had higher than 1 expression ratios at 24 h (Supplementary Fig. S4b). In both conditions, most of the *gli* genes and *gtmA* had higher than 1 expression ratios in more than a single timepoint (Supplementary Fig. S4a and S4b). However, both the wild-type and Δ*pexG* mutants did not produce GT under non-inducing conditions but as expected Δ*pexG* had much higher GT production than the wild-type under inducing conditions (Supplementary Fig. S4c). Taken together, these results suggest that although PexG can affect *gli* genes and *gtmA* mRNA levels accumulation under both non-inducing and inducing conditions, the increased GT production in the Δ*pexG* only occurs under inducing conditions.

These results strongly indicate that PexE and PexG are involved in *A. fumigatus* GT production and self-protection, and virulence, and also affect the production of other SMs. Our results also indicate *A. fumigatus* and *A. nidulans* PexE homologs have different functions in the GT self-protection since *A. fumigatus* Δ*pexE* is much more sensitive to GT than *A. nidulans* Δ*pexE* strain.

## Discussion

We can envisage the following main non-exclusive mechanisms of SM self-defense in fungi (adapted from[1,46]): (i) secretion of toxic compounds in the extracellular milieu, (ii) sequestration of toxic compounds into vacuoles, (iii) modification of toxic compounds into inactive forms, and (iv) increased copy number of SM targets. GT is an important SM for *A. fumigatus* virulence and pathogenicity. GT production induces the modulation of several metabolic pathways, such as sulfur assimilation and transsulfurylation pathways, oxidative stress defenses, methylation, and iron metabolism aiming to cope with GT biosynthesis and self-protection (for reviews, see[12,15,22,47,48]). The two main mechanisms of GT detoxification are transport through the GliA, a MFS transporter and GT modification through the action of two enzymes responsible for GT detoxification, GliT oxidoreductase and GtmA methyltransferase[15]. There was some speculation in the literature if dtGT might be sequestered into intracellular vesicles and converted by an exocytotic mechanism complementary to GliA-mediated efflux[20]. It is technically very challenging to identify the subcellular location of either dtGT, bmGT, or GT. However, it is possible to assess the subcellular localization of the two major players in the conversion of GT-dtGT-bmGT, GliT, and GtmA, in the production of these

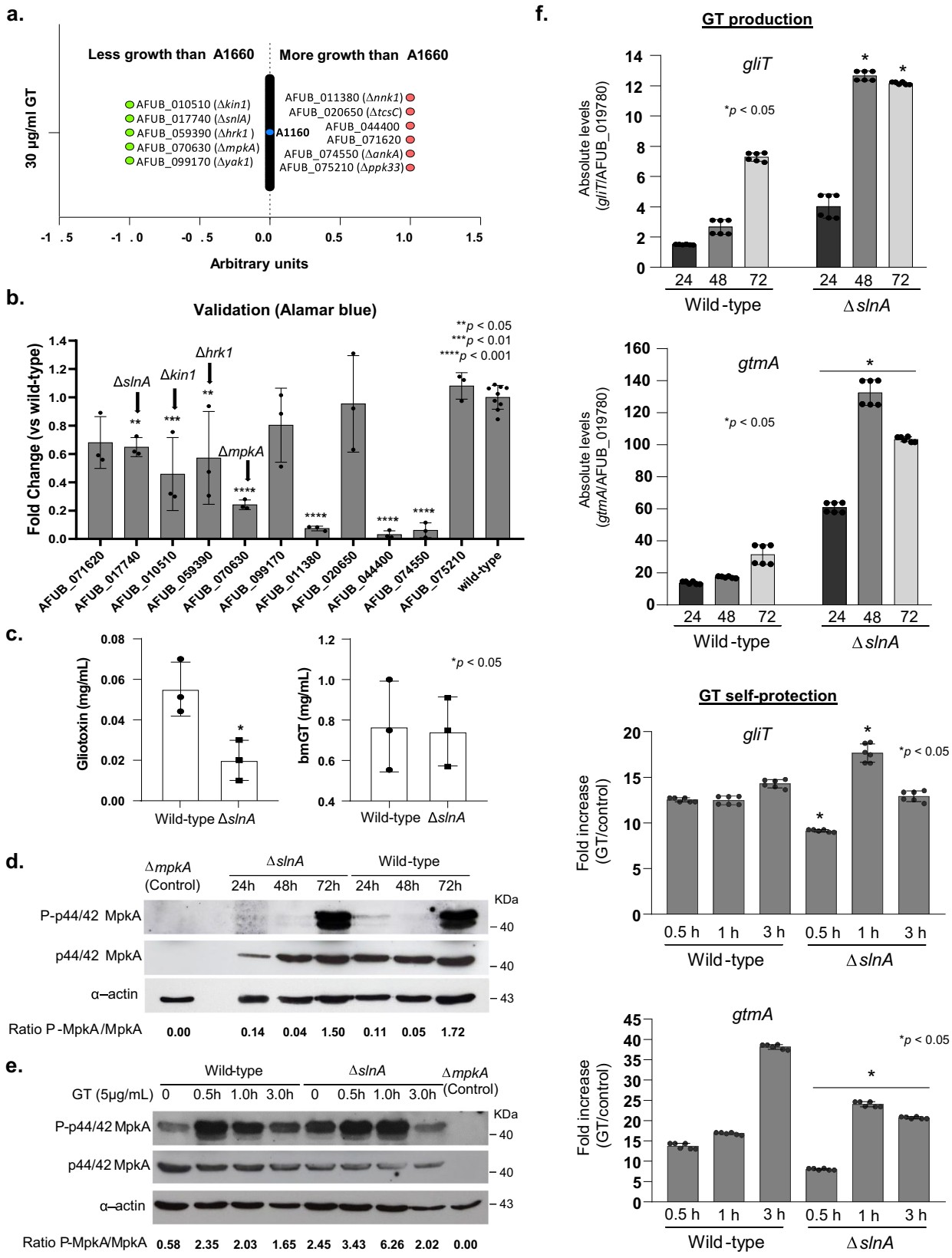

compounds by constructing functional GFP strains. Our work provided several lines of evidence suggesting that part of the cellular supply of dtGT, bmGT, and GT could be stored into vacuoles and/or vesicles during GT production.

Short GT exposure displayed a more complex and diffuse pattern of GliT:GFP and GtmA:GFP subcellular localization in the cytoplasm,

vesicles, endomembraneous compartments, and endoplasmic reticulum. Interestingly, GliT and GtmA are not observed as localized in the vacuoles during self-protection upon short exposure times (like 2 h exposure to 3 μg/mL GT). Longer exposures to the same concentration of GT, such for instance 8 and 16 h, did not allow us to be assertive about a possible colocalization in the vacuoles since the levels of

**Fig. 5 | Screening of the protein kinase null mutant library for GT susceptibility.** **a** Growth screening. **b** Metabolic activity expressed by Alamar blue of *A. fumigatus* grown for 48 h in the absence or presence of 30 μg/mL of GT (*, $p < 0.05$; **, $p < 0.01$, and ***, $p < 0.001$, One-way ANOVA with Dunnett's multiple comparisons test). All the results are the average of three repetitions ± standard deviation. **c** GT and bmGT production in the *A. fumigatus* wild-type and Δ*slnA* strains. The strains were grown in liquid liquid Czapek-dox medium for 3 days at 37 °C. GT and bmGT were extracted from the supernatants and analyzed by LC-MS. Results are the mean of $n = 6$ biological replicates from 3 independent experiments ± SD; (*, $p < 0.05$, *t*-test). **d**, **e** Western blot analysis for GT production and self-protection. Western blot analysis for GT production and self-protection. GliT:GFP and GtmA:GFP were identified by anti-GFP antibody. MpkA-P and total MpkA were identified by using P-p44/42 and p44/42 antibodies while actin by anti-actin antibody. For GT production *A. fumigatus* was grown under the same conditions described in (**c**) while for GT self-protection *A. fumigatus* was grow in liquid minimal medium for 24 h at 37 °C and exposed or not to GT 5 μg/mL for 0.5 to 3 h. **f** RTqPCR for *gliT* and *gtmA* for wild-type and Δ*slnA* strains under GT production and self-protection conditions, as described in (**d**, **e**) of 3 biologically independent experiments (with two technical replicates). Results are the mean values ± SD; (*, $p < 0.05$, *t*-test). Source data are provided as a Source Data file.

GliT:GFP and GtmA:GFP expression were high and massively distributed along the whole cytoplasm. It is possible the small vesicles observed populated by GliT:GFP and GtmA:GFP are derived from the endoplasmic reticulum. Since the fungus is exposed to a high GT concentration, part of the free GT is immediately transported to these vesicles, metabolized or not to dtGT and bmGT, and secreted through vesicles. In both cases, production and self-protection, it remains to be determined how GT is transported to the vesicles and/or vacuoles. Recently, we identified two transporter encoding genes (a MFS transporter, AN1472/AFUA_8G04630, and an ATP-binding cassette transporter, AN7879/AFUA_1G10390) that are RglT-dependent and positively modulated at transcriptional level upon *A. fumigatus* and *A. nidulans* exposure to GT[36]. It remains to be demonstrated if these transporters are located into the vacuole cell membranes and if they could be possible candidates for GT transport to the vacuoles.

Fungal MAPK pathways are essential for the regulation of several cellular processes and different kinds of stress[50–53,56]. The central module for each MAPK has three protein kinases: a MAP kinase kinase kinase (MAPKKK), a MAP kinase kinase (MAPKK), and a MAPK. MAPK cascades are usually stimulated by upstream sensors, such as cell membrane receptors, and its final output is the activation of downstream elements, such as cytoplasmic proteins and transcriptional regulators[50–53]. *A. fumigatus* has four MAPK: MpkA, which mainly regulates cell wall integrity[56]; MpkC and SakA, similar to *Saccharomyces cerevisiae* Hog1, which are involved in the response to oxidative and osmotic stresses[57–59] and MpkB, homologous to yeast Fus3 and involved in melanin production[60,61]. We have demonstrated that GliT and GtmA are physically interacting with the MAPKs SakA, MpkB, and MpkA during GT production and self-protection. The *sakA*, *mpkC*, and double *sakA mpkC* null mutants have comparable GT levels than the wild-type. The Δ*mpkA* mutant is not able to produce either GT or bmGT, and is more sensitive to GT. Interestingly, MpkA affects the GliT and GtmA RNA and protein levels and its subsequent translocation to the vacuoles. Previously, it has been shown that *A. fumigatus* MpkA is also involved in the control of the production of several SMs, such as melanin, pseurotin A, siderophores, and GT[39,61]. However, the mechanism of how MpkA regulates these SMs has not been previously investigated. By using co-IPs, we have validated MpkA interaction with GliT and GtmA but it is yet to be investigated if GliT and GtmA are directly interacting with MpkA or through other MpkA-associated proteins and in this case which residues are phosphorylated by MpkA. We also observed GliT and GtmA physical interactions with other protein kinases and phosphatases, such as calcium/calmodulin-dependent protein kinases, GskA kinase, and PphB and the MAPK-dependent PtcB phosphatases. It remains to be determined the roles played by these enzymes for the regulation of GT production and self-protection.

In a screening performed with a genome-wide collection of *A. fumigatus* non-essential PK mutants, we have also identified other PKs besides MpkA as important for GT self-defense. One of these kinases, the histidine kinase SlnA, was previously extensively characterized in our laboratory and we have not observed striking phenotypes in this mutant apart from a reduction of its radial diameter of about 15 % when compared to the wild-type strain[41]. SlnA is the homolog of *S. cerevisiae* Sln1, a two-component system that controls the branch of the yeast HOG pathway[57,62,63]. The typical organization of a two-component system consists of the following components: (i) a sensor histidine kinase (SHK) that contains an input (or sensor) domain, an HK catalytic domain, and a histidine autophosphorylation site, and (ii) a response regulator (RR) that contains a receiver (REC) domain and an output (or effector) domain[64]. The sensor domain is modified by a stimulus, a histidine close to the HK domain is phosphorylated (or dephosphorylated), and this phosphoryl group is transferred to the REC domain of the RR[64]. We observed that SlnA is important for the modulation of the MpkA phosphorylation during GT self-protection but not GT production, and *slnA* null mutant has lower GT production than the wild-type. These results indicate that SlnA is one of the sensors that activate GT self-protection via MpkA phosphorylation. Since GT production is intimately related to GT self-production it is possible the reduction of GT production in the *slnA* null mutant is due to a reduction of GliT and GtmA activities, and consequently low GT accumulation.

Fungal peroxisomes are specialized in both anabolism and catabolism with very important functions for cell protection by detoxifying and sequestering reactive oxygen species[49,50]. Acetyl CoA generation by β-oxidation of fatty acids and the enzymes isocitrate lyase and malate synthase specific to the glyoxylate cycle for the capture of acetyl CoA for gluconeogenesis or to provide intermediates for the TCA cycle within mitochondria are also present in the peroxisomes[49,50]. Several enzymatic steps for diverse fungal SMs, such penicillin, aflatoxin, and sterigmatocystin, occur within the peroxisomes (for a review, see[50]). We investigated the influence of peroxisomes on GT production and self-protection by characterizing *A. fumigatus pexE* and *pexG* deletion mutants. Interestingly, while Δ*pexE* produces very low levels of GT and bmGT and is very sensitive to GT, the Δ*pexG* mutant is as sensitive to GT as the wild-type and has increased production of GT. These results suggest that both receptors play a role in GT production and self-protection. Although there is increased production of *gli* genes and *gtmA* mRNA levels in the Δ*pexG* mutant in both non-inducing and inducing conditions, the increased GT production in this mutant only occurs in inducing conditions. These results suggest that PexG modulates the production of GT but this modulation only happens through the same signal transduction pathways that coordinate the GT induction. It remains to be investigated which proteins are transported to the peroxisomes via PexG and are involved in the modulation of GT production under GT inducing conditions. What could be the peroxisomal mechanisms influencing GT production and self-protection? We detected increased sensitivity of Δ*pexG to* menadione that could suggest peroxisomes are required for the optimal oxireduction cellular environment necessary for GT production and self-protection. We were not able to assign any other phenotype that could impact GT self-protection in these mutants, such as sulfur source utilization. Interestingly, PexE and PexG are also involved in self-defense in *A. nidulans*, a non GT producer. Our results indicate a role for peroxisomes in GT production and self-defense, and virulence.

In summary, our work demonstrates another mechanism of GT self-protection through a possible GT-dtGT-bmGT storage in the

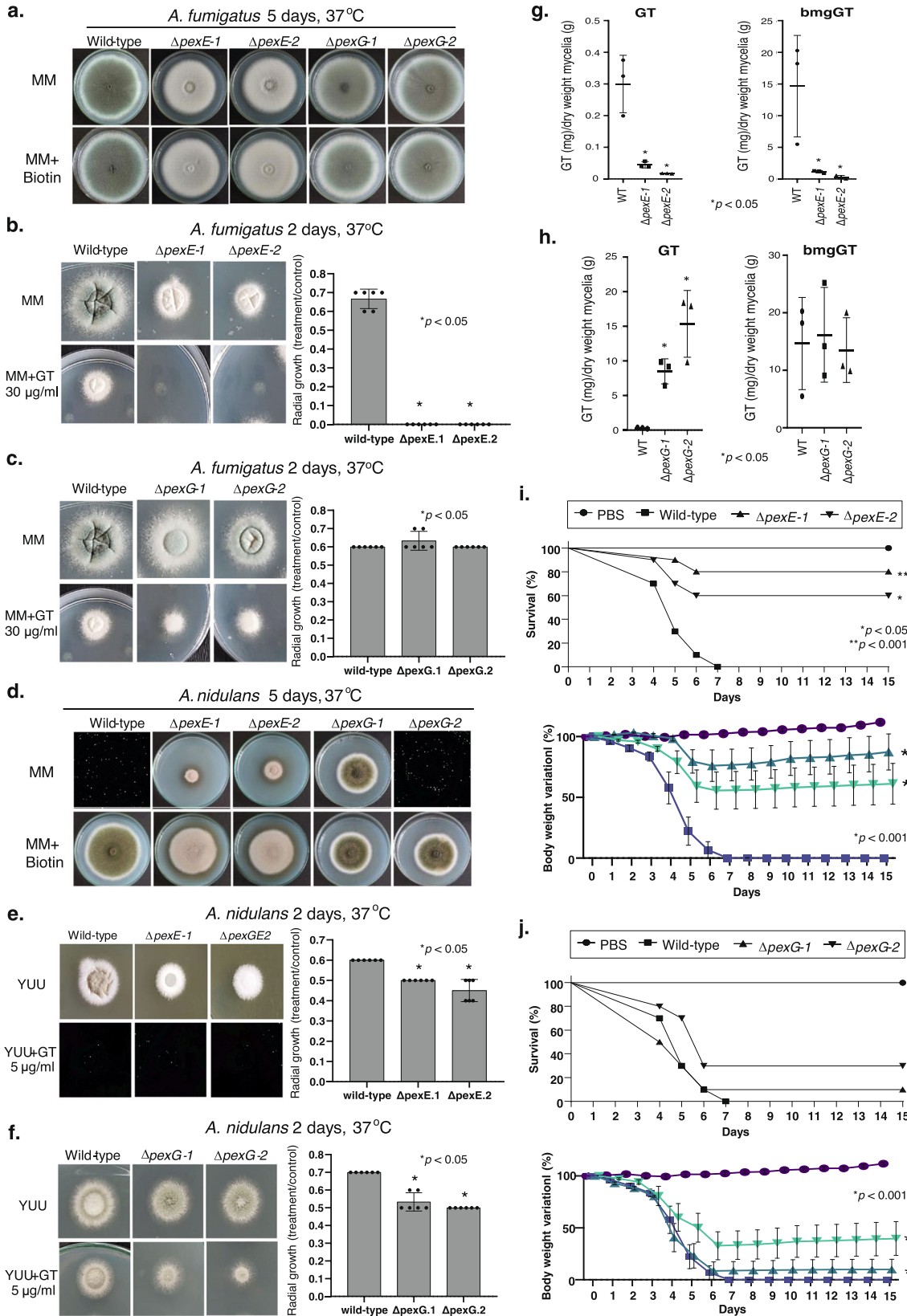

vacuolar system. Surprisingly, we observed a function for peroxisomes in the GT production and self-protection. Further work will focus on the mechanisms how GT is transported into vacuoles, GliT and GtmA are partially localized into the vacuoles, and peroxisomes influence GT production and self-protection.

## Methods

### Ethical statement

The principles that guide our studies are based on the Declaration of Animal Rights ratified by UNESCO on January 27, 1978 in its 8th and 14th articles. All protocols adopted in this study were approved by the

**Fig. 6 | Peroxisomes are required for GT production, self-protection, and virulence. a** The *A. fumigatus* wild-type, Δ*pexE*, and Δ*pexG* strains were grown for 5 days at 37 °C on MM and MM+biotin. **b**, **c** The *A. fumigatus* wild-type, Δ*pexE*, and Δ*pexG* strains were grown for 2 days at 37 °C on MM and MM + 30 μg/ml of GT. The radial growth results are the mean of 3 independent experiments ± SD, *n* = 6 biological replicates (*, *p* < 0.05, *t*-test) **d** The *A. nidulans* wild-type, Δ*pexE*, and Δ*pexG* strains were grown for 5 days at 37 °C on MM and MM+biotin. **e**, **f** The *A. nidulans* wild-type, Δ*pexE*, and Δ*pexG* strains were grown for 2 days at 37°C on YUU and YUU + 5 μg/ml of GT. Results are the mean of 3 independent experiments ± SD, *n* = 6 biological replicates (*, *p* < 0.05, *t*-test). **g**, **h** GT and bmGT production in the *A. fumigatus* wild-type, Δ*pexE*, and Δ*pexG* strains. The strains were grown in liquid Czapek-dox medium for 3 days at 37 °C. GT and bmGT were extracted from the supernatants and analyzed by mass spectrometry. Results are the mean of *n* = 6 biological replicates from 3 independent experiments ± SD; (*, *p* < 0.05, *t*-test). **i**, **j** Kaplan-Meier survival curves showing percentages of immunosuppressed mice infected intranasally with *A. fumigatus* strain (*n* = 10 mice/strain) and percentage of variation in the body weight. Phosphate buffered saline (PBS) was administered in a negative control group (*n* = 10). The indicated *p*-values determined with the use of the Logrank (Mantel-Cox test) and Gehan-Breslow-Wilcoxon test comparing the Δ*pexE*, and Δ*pexG* mutants with the wild-type strain, (*, *p* < 0.05 and **, *p* < 0.001). In the percentage of body weight variation, Δ*pexE* mutants are compared to the wild-type while Δ*pexG* mutants are compared to the PBS negative control (*, *p* < 0.001, One-way ANOVA with Tukey's multiple comparisons test). Data represent the cumulative data of 2 separate experiments. Source data are provided as a Source Data file.

**Table 2 | Secondary metabolites (SMs) produced by *A. fumigatus* wild-type, Δ*pexE*, and Δ*pexG***

| SMs[*] | Wild-type | Δ*pexE* | *p* < 0.05 | Δ*pexG* | *p* < 0.05 |
|---|---|---|---|---|---|
| Brevianamide F | 3.1 ± 0.2e7 | 2.3 ± 1.6e6 | 0.00 | 8.3 ± 0.5e6 | 0.00 |
| Cycloprostatin A | 0.00 ± 0.00 | 3.0 ± 0.8e6 | 0.01 | 5.7 ± 1.9e6 | 0.02 |
| Cyclotryprostatin A | 2.9 ± 1.9e6 | 0.00 ± 0.00 | 0.02 | 0.00 ± 0.00 | 0.02 |
| Cyclotryprostatin C | 0.00 ± 0.00 | 0.00 ± 0.00 | ND | 2.9 ± 0.9e8 | 0.01 |
| Demethoxyfumitremorgin C | 4.8 ± 2.1e6 | 2.8 ± 1.1e6 | 0.12 | 8.5 ± 0.7e7 | 0.00 |
| Fumagillin | 4.4 ± 1.0e7 | 0.00 ± 0.00 | 0.01 | 8.7 ± 0.9e7 | 0.00 |
| Fumigaclavine C | 0.00 ± 0.00 | 2.4 ± 0.4e6 | 0.00 | 9.3 ± 2.3e6 | 0.01 |
| Fumiquinazoline A or B | 2.3 ± 0.9e6 | 6.2 ± 4.8e6 | 0.15 | 1.5 ± 0.5e8 | 0.02 |
| Fumiquinazoline E | 3.2 ± 1.5e6 | 1.7 ± 0.7e6 | 0.12 | 8.8 ± 3.9e6 | 0.06 |
| Fumiquinazoline F or G | 1.7 ± 0.4e7 | 3.2 ± 2.8e7 | 0.23 | 5.8 ± 0.5e8 | 0.00 |
| Fumitremorgin A | 0.00 ± 0.00 | 2.1 ± 1.3e5 | 0.06 | 0.00 ± 0.00 | ND |
| Fumitremorgin B | 0.00 ± 0.00 | 0.00 ± 0.00 | ND | 2.3 ± 0.6e6 | 0.01 |
| Fumitremorgin C | 1.4 ± 0.6e7 | 1.7 ± 1.1e7 | 0.03 | 6.6 ± 1.8e8 | 0.01 |
| Fungisporin | 1.3 ± 0.2e7 | 0.00 ± 0.00 | 0.00 | 7.0 ± 04e6 | 0.05 |
| Pseurotin A | 1.5 ± 0.1e7 | 5.6 ± 5.2e6 | 0.05 | 1.8 ± 0.3e7 | 0.10 |
| Pseurotin D | 0.00 ± 0.00 | 5.7 ± 1.6e6 | 0.01 | 0.00 ± 0.00 | ND |
| Pseurotin F2 | 0.00 ± 0.00 | 2.4 ± 2.4e6 | 0.11 | 3.1 ± 0.7e6 | 0.01 |
| Pyripyropene A | 4.6 ± 2.3e6 | 1.4 ± 0.5e6 | 0.06 | 2.8 ± 1.7e6 | 0.17 |
| Pyripyropene E | 0.00 ± 0.00 | 1.6 ± 1.3e6 | 0.08 | 1.0 ± 0.4e7 | 0.03 |
| Pyripyropene F | 0.00 ± 0.00 | 0.00 ± 0.00 | ND | 1.5 ± 0.8e6 | 0.04 |
| Pyripyropene N | 4.8 ± 0.2e6 | 1.1 ± 0.5e7 | 0.08 | 3.3 ± 1.4e6 | 0.11 |
| Spirotryprostatin A | 4.6 ± 1.8e7 | 2.0 ± 1.1e7 | 0.07 | 2.4 ± 0.3e8 | 0.00 |
| Trypostatin A | 1.2 ± 0.4e7 | 2.6 ± 1.7e6 | 0.02 | 9.7 ± 1.1e6 | 0.26 |
| Trypostatin B | 6.2 ± 1.3e6 | 6.3 ± 6.5e6 | 0.49 | 2.9 ± 0.8e8 | 0.01 |

[*]The results represent the average of the areas of the chromatograms of three independent biological repetitions ± SD. Statistical analysis was performed by using *t*-test (*p* < 0.05) and comparing the mutants versus the wild-type. *ND* not determined.

local ethics committee for animal experiments from the University of São Paulo, Campus of Ribeirão Preto (Permit Number: 08.1.1277.53.6; Studies on the interaction of *Aspergillus fumigatus* with animals). Groups of five animals were housed in individually ventilated cages and were cared for in strict accordance with the principles outlined by the Brazilian College of Animal Experimentation (COBEA) and Guiding Principles for Research Involving Animals and Human Beings, American Physiological Society. All efforts were made to minimize suffering. Animals were clinically monitored at least twice daily and humanely sacrificed if moribund (defined by lethargy, dyspnea, hypothermia, and weight loss). All stressed animals were sacrificed by cervical dislocation.

## Strains and media
All strains used in this study are listed in Supplementary Table S2. Strains were grown at 37 °C. Conidia of *A. fumigatus* and *A. nidulans* were grown on complete medium (YG) [2% (w/v) glucose, 0.5% (w/v)

yeast extract, trace elements] or minimal media (MM) [1% (w/v) glucose, nitrate salts, trace elements, pH 6.5]. Solid YG and MM were the same as described above with the addition of 2% (w/v) agar. When necessary, uridine and uracil (1.2 g/L) were added (YAG + UU = YUU or MM + UU = MMUU). Trace elements, vitamins, and nitrate salt compositions were as described[65]. Gliotoxin production was induced by growing the strains in Czapek-Dox (http://himedialabs.com/TD/M076.pdf) broth. For phenotypic characterization, plates were inoculated with $10^4$ spores per strain and left to grow for 120 h at 37 °C. Radial growth experiments were expressed as ratios, dividing colony radial diameter of growth in the stress condition by colony radial diameter in the control (no stress) condition.

## Microscopy
For microscopic analyzes of GFP fluorescence under gliotoxin self-protection condition, the strains were grown on coverslips in 4 mL of MM for 24 h at 37 °C. After incubation, the coverslips with adherent

germlings were left untreated or treated with for gliotoxin (Sigma Aldrich, St. Louis, USA) at a final concentration of 3ug/mL for 2 h. Gliotoxin production was induced by growing the strains in Czapek-Dox broth for 24 h. CMAC (CellTracker Blue CMAC - Molecular Probes) staining of vacuoles was performed by adding 10 μM CMAC dye to the cultures for 15 min, at 37 °C; and ER-Tracker Blue-White DPX (https://www.thermofisher.com/order/catalog/product/E12353), 10 min, was used to stain the ER. Subsequent to the staining, the coverslips were rinsed with PBS and mounted for examination. In the GT production assay, germlings were assigned as positively present in the vacuoles if they have concomitant localization of GliT:GFP or GtmA:GFP in the same vacuole stained by CMC in the same germling. In the GT self-protection assay, germlings were assigned as positively present in the vacuoles, endoplasmic reticulum, vesicles, or endomembranes if at least one of these structures was co-localizing with GliT:GFP or GtmA:GFP in the same germling. Slides were visualized on a Carl Zeiss Observer Z1 fluorescence microscope using the excitation wavelength of 450 to 490 nm, and emission wavelength of 500 to 550 nm. DIC (differential interference contrast) images and fluorescent images were captured with an AxioCam camera (Carl Zeiss) and processed using AxioVision software (version 4.8).

### Gliotoxin and bmGT extraction and analysis by High Performance Liquid Chromatography (HPLC) and liquid chromatography-mass spectrometry (LC-MS)

The strains ΔmpkA, ΔmpkB, A1163, ΔsakA ΔmpkC, ΔmpkC, ΔsakA, ΔpexE, ΔpexG, GtmA:GFP, GliT:GFP, vesicles of GtmA:GFP, GliT:GFP and WT (1 mL) were grown in Czapek-Dox broth for 72 h as this medium has previously been shown to result in detectable gliotoxin levels in culture supernatants[20]. For extraction, the cultures were submitted to liquid-liquid partition with 15 mL, 300 mL and 300 μL of chloroform, respectively, for three times. Organic fractions were washed with a saturated solution of NaCl and dried with anhydrous $Na_2SO_4$. The suspensions were filtered and concentrated under vacuum. The instrumentation for the LC-MS used was the Shimadzu Nexera XR LC-20AD (Kyoto, Japan) chromatography model, consisting of CBM-20A control, SPD-M20A DAD and ELSD-LTII evaporative light scattering detectors, Nexera SIL-20A auto injector, CTO-20A oven, using reversed phase C18 column (Ascentis, 2.7 μm, 100 × 4.6 mm, 35 °C) with gradient of aqueous (0.1% acetic acid) and acetonitrile (10% to 100% of acetonitrile) for 35 min. LabSolutions software (Shimadzu Corporation, Kyoto, Japan) was used for data acquisition and analysis.

### Determination of SMs produced by the wild-type and *pex* mutants

The samples used in the Liquid Chromatography-High Resolution Mass Spectrometry (LC-HRMS) analyzes were prepared from the extracts obtained for the different *A. fumigatus* strains grown for 5 days at 37 °C in liquid MM. Each sample (supernatant) was prepared from 100 mg of extract, which were dissolved in HPLC grade methanol. SMs were extracted for 90 min in an ultrasonic bath at room temperature. Then, filtration was performed through a 0.22 μm sterile syringe filter and transferred to 2 ml HPLC vials.

Analysis LC-HRMS were performed in an UHPLC-MS/MS - Thermo Q Exactive Orbitrap Mass Spectrometers (MS) - Dionex UltiMate 3000 RSLCnano System, operating in positive mode. LC analyzes were performed in a C18 (100 mm × 2.1 mm × 2.6 mm; Thermo) column. The gradient was initiated with 5% B mobile phase (0.1% formic acid in ACN), which was maintained for 5 min, followed by linearly increasing to 40% B within 5 min, then increased to 45% B in 2 min, then up to 98% B in 18 min and held for 2 min. Finally, the phase was changed to 95% A (0.1% formic acid) in 2 min and maintained until 24 min. Mass spectrometry analysis was performed in full scan, with a scan range from *m/z* 100 to 1500 Da. MS/MS fragmentation spectra were acquired from the five most intense ions per scan. The injection volume was 5 μL.

The SMs annotations were classified according to[66]. **Annotation Level I**, confirmation of structure by comparing the MS/MS profile with reference standards; **Annotation Level II**, through the comparison of MS/MS fragments with correspondence spectra present in the GNPS and other databases; **Annotation Level III**, candidates who presented structural evidence through in silico generated MS and MS/MS fragments; **Annotation Level IV**, annotated SMs in which it was possible to determine the molecular formula unequivocally through the information contained in the LC-HRMS analyzes. In this study the simulated spectra used were generated by SIRIUS 5.6.3[67] and the input data were automatically compared and classified against databases present in SIRIUS 5.6.3[68].

Finally, the areas corresponding to the annotated ions were calculated manually, as a way of determining their production in the different mutants of *A. fumigatus*, using the software Thermo Xcalibur 3.0.63 (Copyright 1988-2013 Thermo Fisher Scientific Inc.).

### Generation of A. nidulans and A. fumigatus mutants

All gene replacement cassettes were constructed by "in vivo" recombination in *S. cerevisiae*, as previously described by[69,70]. For construction of *A. nidulans and A. fumigatus pex5* and *pex7* null mutants, approximately 1.0 kb from each 5-UTR and 3-UTR flanking region of the targeted ORFs regions were selected for primer design (P1 to P16). The primers *gene*_pRS426_5UTR_fw and *gene*_pRS426_3UTR_rv contained a short homologous sequence to the MCS of the plasmid pRS426. Both the 5- and 3-UTR fragments were PCR-amplified from the genomic DNA of *A. nidulans* AGB551 strain or *A. fumigatus* CEA17, pyrG⁻ strain. The *pyrG* gene placed within the cassette as a prototrophic marker was amplified from pCDA21[71,(76)] plasmid using the primers P17/P18. The cassette was PCR-amplified from these plasmids utilizing TaKaRa Ex Taq™ DNA Polymerase (Clontech Takara Bio) and used for *A. nidulans* and *A. fumigatus* transformation. Southern blot analysis was performed to confirm the deletions (Supplementary Fig. S5). To generate the MpkA:linker-3xHA-trpC-pyrG fusion fragment, a 2.6 Kb portion of DNA consisting of the 5-UTR region and mpkA ORF, along with a 1 Kb segment of DNA consisting of the 3-UTR flanking region were amplified with primers P19/P20 and P21/P22, respectively, from CEA17 gDNA. The 2.7 kb 3xHA - trpC - pyrG fusion was amplified with primers OZG916/OZG964 from the pOB430 plasmid. For the cassettes GtmA:linker-GFP-trpC-prtA and GliT:linker-GFP-trpC-prtA the fragments 5-UTR + ORF and 3-UTR (1Kb) were also PCR amplified from CEA17 gDNA with primers P23 to P30. The linker-GFP-trpC fragment was amplified from the pOB435 plasmid with primers P31/P32, and the prtA gene was amplified from the plasmid pPTRI with primers P33/P34. Cassettes were generated by transforming each fragment along with the plasmid pRS426 cut with BamHI/EcoRI into the *S. cerevisiae* strain. The DNA plasmid of the transforming bacteria was extracted, cassettes were PCR-amplified from these plasmids utilizing TaKaRa Ex Taq™ DNA Polymerase, which were subsequently transformed into the background of the CEA17, pyrG⁻ (for construction of the MpkA:HA::-pyrG) and MpkA:HA::pyrG (for construction of the double-tagged strains: MpkA:HA::pyrG, GtmA:GFP::prtA or MpkA:HA::pyrG, GliT:GFP::prtA). Mutants were selected on MM or MM supplemented with 1 μg/mL pyrithiamine and confirmed with PCR. Primer sequences are listed in Supplementary Table S3.

### Western blot analysis

Total cellular protein extractions were carried out as described[71], and quantified using Bradford reagent (Bio-Rad), according to manufacturer's instructions. Fifty μg of protein from each sample were resolved in a 12% (w/v) SDS−PAGE and transferred to polyvinylidene difluoride (PVDF) membranes (Merck Millipore). GFP-tagged strains were detected using 1:5,000 dilution of the mouse monoclonal GFP

antibody (Santa Cruz Biotechnology) and secondary antibody anti-mouse IgG HRP conjugate (Cell Signaling Technology), at 1:5,000 dilution. For the HA-tagged proteins detection, a mouse monoclonal anti-HA antibody (Sigma) was used at 1:5,000 dilution as a primary antibody, followed by the same anti-mouse IgG HRP conjugate as a secondary antibody. The phosphorylated fractions of the MAP kinase, MpkA, were examined using anti-phospho p44/42 MAPK antibody (Cell Signaling Technologies) following the manufacturer's instructions using a 1:10,000 dilution. The primary antibody was detected using an HRP-conjugated secondary antibody raised in rabbit (Sigma). Chemoluminescent detection was achieved using an ECL Prime Western Blot detection kit (GE HealthCare). To detect these signals on blotted membranes, the ECL Prime Western Blotting Detection System (GE Helthcare, Little Chalfont, UK) and LAS1000 (FUJIFILM, Tokyo, Japan) were used.

### Murine model of pulmonary aspergillosis

Wild-type BALB/c female mice, body weight 20 to 22 g, aged 8-9 weeks, were kept in the Animal Facility of the Laboratory of Molecular Biology of the School of Pharmaceutical Sciences of Ribeirão Preto, University of São Paulo (FCFRP/USP). Cages were well ventilated, softly lit and subjected to 12:12 hs light-dark cycle. The relative humidity was kept at 40 to 60%. Mouse rooms and cages were kept at a temperature of 22ºC. The mice were given food and water ad libitum throughout the experiments. The procedures adopted in this study were performed in accordance with the principles of ethics in animal research and were approved by the Committee on Ethics in the Use of Animals (CEUA) of the FCFRP/USP (Permit Number: 08.1.1277.53.6; Studies on the interaction of *Aspergillus fumigatus* with animals) from the University of São Paulo, Campus of Ribeirão Preto.

Mice (10 per treatment) were immunosuppressed with cyclophosphamide (150 mg per kg of body weight), which was administered intraperitoneally on days −4, −1, and 2, prior to and post infection. Hydrocortisonacetate (200 mg/ kg body weight) was injected subcutaneously on day −3. *A. fumigatus* strains were grown on YAG for 2 days prior to infection. Fresh conidia were harvested in PBS and filtered through a Miracloth (Calbiochem). Conidial suspensions were spun for 5 min at 3000 x *g*, washed three times with PBS, counted using a hemocytometer, and resuspended at a concentration of $5.0 \times 10^6$ conidia/ ml. The viability of the administered inoculum was determined by incubating a serial dilution of the conidia on YAG medium, at 37 °C. Mice were anesthetized by halothane inhalation and infected by intranasal instillation of $1.0 \times 10^5$ conidia in 20 μl of PBS. As a negative control, a group of 10 mice received PBS only. Mice were weighted every 24 h from the day of infection and visually inspected twice daily. For body weight variation (%), all weights are calculated in relation to day 0 (day of infection). The formula is described in the Source data. The statistical significance of comparative survival values was calculated by the Prism statistical analysis package by using Log-rank (Mantel-Cox) Test and Gehan-Brestow-Wilcoxon tests.

### GFP or HA-tag protein purification and identification by LC-MS/MS

To precipitate GFP/HA-tag labeled strains, protein crude extracts were prepared from cultures grown for 24, 48 and 72 h in Czapek Dox or in MM followed by gliotoxin treatment (5 μg / mL). Crude protein extracts from mycelia were obtained by extraction from ground mycelia with B250 buffer (250 mM NaCl, 100 mM Tris–HCl pH 7.5, 10% glycerol, 1 mM EDTA and 0.1% NP-0.4) supplemented with 1.5 mL/L 1 M DTT, 2 tablets/100 mL complete-mini protease inhibitor cocktail EDTA-free (Roche), 3 mL/L 0.5 M Benzamidine, 10 ml/L phosphatase inhibitors 100× (10 M NaF, 5 M Na Vanadate, 8 M β- glycerol phosphate), and 10 mL/L 100 mM PMSF. Extracts were centrifuged at 13,000 *g* for 20 min at 4 °C, and the supernatant was collected into a new eppendorf. The same amount of protein for each sample was added to 20 μl of Magnetics GFP-trap beads or to Dynabeads Protein A (Thermo Fisher Scientific) previously incubated with monoclonal anti-HA antibody (Sigma). Cell extracts and beads were incubated with shaking at 4 °C for 4 h. After incubation, the magnetics beads were collected using magnetic hack and washed according to manufacturer's instructions. To release the proteins from the beads, samples were incubated with sample buffer and boiled at 98 °C for 5 min. Western Blot assays were carried out as described above. For the LC-MS/MS identification, the washed beads were re-suspended in 50 mM ammonium bicarbonate solution. Proteins were reduced with 10 mM DTT (DL-Dithiothreitol–Sigma-Aldrich) for 20 min at 56 °C, alkylated with 40 mM iodoacetamide (Sigma-Aldrich) for 15 min at room temperature in the dark, and digested with trypsin (Promega) in the ratio 1:50 (μg trypsin/μg protein) at 37 °C overnight. The digestion was stopped by the addition of trifluoroacetic acid (TFA) to reach a final concentration of 1% and then the sample was desalted with C18 columns (StageTips were used; they are centrifugeable, compatible with automates for high throughput screening, and useful for biomolecular applications of peptide desalting and proteomics, ZipTip, EMD Millipore, catalog number ZTC18S960). Interaction partners of GliT and GtmA were analyzed in at least three biological replicates filtered for unspecific peptides identified with AFUXX (untagged control strain). Peptides were run in LC-MS/MS and analyzed as described in detail in Ref. 72.

### RNA extraction, cDNA synthesis and RT-qPCR

All experiments were carried out in biological triplicates, and conidia ($10^7$) were inoculated in the liquid culture medium [Czapek Dox ou MM (with or without gliotoxin treatment)]. For total RNA isolation, mycelia were ground in liquid nitrogen and total RNA was extracted using TRIzol (Invitrogen), treated with RQ1 RNase-free DNase I (Promega), and purified using the RNAeasy kit (Qiagen) according to the manufacturer's instructions. RNA was quantified using a NanoDrop. For RT-qPCR, the RNA was reverse transcribed to cDNA using the ImProm-II reverse transcription system (Promega) according to the manufacturer's instructions, and the synthesized cDNA was used for real-time analysis using the SYBR green PCR master mix kit (Applied Biosystems) in the ABI 7500 Fast real-time PCR system (Applied Biosystems, Foster City, CA, USA). Sybr Primer sequences are listed in Supplementary Table S3. Afu2g02680 (Putative matrix AAA protease) gene was used as normalizers[36].

### Statistical analysis

Grouped column plots with standard deviation error bars were used for representations of data. For comparisons with data from wild-type or control conditions, we performed one-tailed, paired *t* tests or one-way analysis of variance (ANOVA). All statistical analyzes and graphics building were performed by using GraphPad Prism 5.00 (GraphPad Software).

### Reporting summary

Further information on research design is available in the Nature Portfolio Reporting Summary linked to this article.

## Data availability

All the data are available as supplementary tables and figures. Proteomic data was deposited at ProteomeExchange (https://www.proteomexchange.org) under the accession number PXD041133 (https://massive.ucsd.edu/ProteoSAFe/dataset.jsp?task=fedcc33de1ac459282153146d7585b33). Extracts from *A. fumigatus* mutants were identified using the GNPS database (http://gnps.ucsd.edu). The GNPS Feature-based Molecular Networking (FBMN) analysis was performed following the protocol already established on the website (https://ccms-ucsd.github.io/GNPSDocumentation/featurebasedmolecularnetworking/). MS/MS spectra were selected

with only the top five fragmentation ions in the ± 5 ppm window. The mass tolerance of precursor ions and MS/MS fragment ions were adjusted to 0.02 Da in both cases. The spectral libraries used for this study were pre-established according to the features table generated by MZMine 3 (http://mzmine.github.io/). Correspondences between network spectra and libraries were filtered to show values greater than 0.7 (Cosine Score). The results obtained for this study are available at: https://gnps.ucsd.edu/ProteoSAFe/status.jsp?task=ea7a4f8507f5433db1a6bc50fb9ca67a. The SMs data was deposited at MassIVE, Center for Computational Mass Spectrometry, full Member of the Proteome Xchange consortium. To access the data, click in the link: https://doi.org/10.25345/C5K35MQ4H. The access number MassIVE: MSV000092041. Source data are provided with this paper.

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

## Acknowledgements

We thank the Fundação de Amparo à Pesquisa do Estado de São Paulo (FAPESP) grants numbers 2016/07870-9 and 2021/04977-5 (G.H.G.) and the Conselho Nacional de Desenvolvimento Científico e Tecnológico (CNPq) grant numbers 301058/2019-9 and 404735/2018-5 (G.H.G.), both from Brazil, and the National Institutes of Health/National Institute of Allergy and Infectious Diseases grant R01AI153356 (GHG) from the USA. A.M.T. was funded by a John and Pat Hume PhD Scholarship from Maynooth University. Protein mass spectrometry facilities were funded by a competitive award from Science Foundation Ireland (12/RI/2346 (3)) to Professor Sean Doyle (Maynooth University). GFP-tagged GliT and GtmA *Aspergillus fumigatus* strains were a kind gift from Professor Sean Doyle (Maynooth University, Ireland). OB and OSB were funded by Science Foundation Ireland grants, 21/FFP-P/10146, and 21/PATH-S/9444, respectively.

## Author contributions

P.A.C., C.F.P., T.F.R., C.V., N.V.R., C.M., I.L.F.M., A.M.T., O.S.B., F.E., and J.C.J.B. performed most of the experiments. M.B., A.B.F., O.B., I.M., T.F., and M.T.P. provided materials and recommended technical approaches for the work. G.H.G. analyzed the data, wrote the manuscript, and coordinated all the work. All the authors read and edited the manuscript.

## Competing interests

The authors declare no competing interests.
