## [Peer Review File · Nature Communications]

Aspergillus fumigatus mitogen-activated protein kinase MpkA is involved in gliotoxin production and self-protectionREVIEWER COMMENTS

Reviewer #1 (Remarks to the Author):

The study by Alves de Castro et al., investigates the role and interactions of GT and its partners in protection and production. The study is informative and interesting with important findings. The concept for host compartmentalization of toxins is well studied and applicable to diverse biological systems, including plants and response to fungal mycotoxins and therefore, the findings may have relevance in other areas. Some statements made need to be clarified (as outlined in revisions) and statistical robustness and descriptions are lacking.

Specific

Line 72: change to ', producing GT as an important...'

Line 90: check that 'kappa' is used for NFkB

Line 99: add comma 'dtGT, attenuating GT...'

Line 138: 'Not only is MpkA involved in...'

Line 139: revise for clarity

Line 154: watch consistency for labeling hours (e.g., hr, hrs, h, hs, hours). Adapt throughout. Also use consistent labeling for 'minutes' or 'min'

Line 185: end parentheses needed

Line 203: 'introduce'

Line 218: missing closed square bracket

Line 267: suggest rephrasing '133 proteins that interact with...' as identification from immunoprecipitation does not imply direct interactions but indirect or complexed proteins may be identified. 'Associate with' may be more appropriate until interactions are directly confirmed.

Line 265: what does 'subtraction' refer to? Were background intensity values subtracted from protein intensity values or were background protein IDs removed from further analysis?

Line 272: were background proteins used for significance calculations (i.e., abundance changes considered for background proteins or presence/absence)?

Line 283: remove duplicate 'including'

Line 340: proteins will be produced; genes will show changes in expression. Suggest revising use of 'expression' to describe production of the protein.

Line 357: does MpkA phosphorylate

Line 337: what is meant by 'MpkA phosphorylation is increased...'? Frequency of phosphorylation or changes in abundance of phosphorylated proteins? If the latter, are phosphorylation-induced abundance changes normalized against changes in the total proteome between the tested conditions? Also relevant for line 387.

Line 392: check units for GT

Line 628: samples transferred to 2 mL HPLC vials (line 617) but 5 mL was injected into the mass spectrometer. Please clarify.

Line 714: how many mice?

Line 768: define Stage Tips (STAGE tips) and add reference

Methods: a P-value for significance is reported in the Results but no reference to data analysis, bioinformatics, or statistical testing is included in the Methods. This needs to be revised.

Line 1140: YUU is mentioned but not included elsewhere in the manuscript. Needs to be revised.

Table 1: revise use of red and green given challenges in distinguishing colors for readers.

Table 2: suggest revising title unless this is based on fulling confirmed interacting proteins

Figures: for microscopy images, how many images were taken and used for the analysis?

Figures: for mass spectrometry information, if statistical testing for p-value determination was used, why are the analyses all qualitative and based on Venn diagrams and protein ID counts?

Reviewer #3 (Remarks to the Author):

This is an interesting work exploring the intracellular detoxification pathways that protect an important human fungal pathogen, *Aspergillus fumigatus*, from toxicity induced during production of the model mycotoxin gliotoxin. The authors generated transgenic strains to study intracellular

trafficking and distribution of two self-protective GT detoxification enzymes encoding for an oxidoreductase and a methyltransferase, GliT and GtmA respectively. They performed a significant amount of work using state of art molecular biology and biochemical tools to identify an important role of peroxisomes in GT detoxification, and an important role of MAP kinase signaling in GT production and GliT and GtmA trafficking. Overall, the work lacks sufficient mechanistic insight on the molecular interaction of GT with the detoxification enzymes of interest and the cellular organelles that are involved in this process.

In particular, a direct role of peroxisomes in intracellular trafficking and activity of GliT and GtmA is not shown (the transgenic molecules fail to colocalize with peroxisomes)

The protective activity of peroxisomes against GT is not explored in detail

The mechanistic link between peroxisomes and MAP kinases is not shown

The precise mechanism of intracellular trafficking of the two self-protective enzymes is not shown and the microscopy methods that are utilized lack sufficient precision to capture these processes (e.g., need for confocal and super-resolution microscopy approaches)

The term "organelle" that is frequently used to describe localization of the enzymes is not appropriate. The authors even fail to use specialized markers to describe ER. Speculation for a non-canonical pathway of secretion of the enzymes is mentioned in the text. However, this has not been experimentally shown.

Reviewer #4 (Remarks to the Author):

This study investigates the controlled production of mycotoxin gliotoxin, a toxin produced by the pathogenic filamentous fungus, *Aspergillus fumigatus*. Mycotoxin gliotoxin (MG) is important for virulence, making this an important pathway to understand. The authors investigate multiple aspects of the cell biology of MG production, which is an important research direction for the field. However, in general the imaging experiments are not quantified, something that is unacceptable for publication as only representative images are shown. Controls are missing in multiple experiments and parts of the paper are confusing. Overall, this read like many different stories thrown into a single manuscript with not enough rigor in many places. This is a very interesting topic, but unfortunately, the data shown do not convince me of the authors conclusions.

Specific comments:

- Figures 1 and 2 present localization experiments, but no quantification is performed. This is not acceptable for publication.
- The rationale for turning to peroxisomes is not made clear by the authors and the authors do not provide evidence that links GliT or GtmA to peroxisomes, yet state that in their title. This is an interesting line of research but needs to be addressed with quantification for publication.
- Throughout the localization experiments, important controls are missing. For example, is vacuole autofluorescence an issue?
- The number of hyphae examined is not listed and again quantification is not performed.
- In Figure 2 the co-localization with FM4-64 and GliT is hard to see. Are the GtmA-GFP puncta dynamic? The hypha in Fig 2F second row (+GT) looks damaged compared to the control and the right hypha.
- In Figure 3 the authors identify GliF and GliN in purified vesicles. Is it possible to localize both proteins with GFP in cells under GT production and self-protection conditions?
- L215 is confusing since the authors only mention in the discussion that GtmA:GFP doesn't co-localize with mRFT:PTS1. It would be useful to include this result in the results section.
- In Figure 4 the authors used a concentration of 30µg/ml GT for the inhibitor study compared to 3µg/ml for the microscopic localization of GliT & GtmA (Figure 2.). Is there a specific reason why? In addition, the *A. nidulans* samples were treated with 5µg/ml. Why the change in concentration?
- Can the authors comment on why the *A. fumigatus* pexE mutant is so sensitive to GT compared to the *A. nidulans* mutant?
- As the *A. fumigatus* pexG mutant shows a 25 to 50 fold increase in GT production - does it have better detoxification mechanisms? Would it make sense to localize GliT and GtmA in the mutant background? It might be interesting to see if the ΔpexG mutant also produces GT in non-inducing

conditions.

- Figure 6, in B the GT production was measured after 72 h. Given the strong growth defect of the mpkA mutant it might be necessary to extend the incubation time by multiple days.

Typos:

L171: CMCA -> CMAC

L175: Figures 2C and 2D -> Fig 2C-E?

L458: PST1 -> PTS1

Answers to the Reviewers

Reviewer #1:

The study by Alves de Castro et al., investigates the role and interactions of GT and its partners in protection and production. The study is informative and interesting with important findings. The concept for host compartmentalization of toxins is well studied and applicable to diverse biological systems, including plants and response to fungal mycotoxins and therefore, the findings may have relevance in other areas. Some statements made need to be clarified (as outlined in revisions) and statistical robustness and descriptions are lacking.

Answer: We thank very much the reviewer for the comments and suggestions. We follow all your suggestions performing additional experiments and adding quantification of the imaging experiments. We believe these comments and suggestions have dramatically improved the quality of the manuscript.

Specific

Line 72: change to ‘, producing GT as an important...’

Answer: This was changed accordingly.

Line 90: check that ‘kappa’ is used for NFkB

Answer: This was changed accordingly.

Line 99: add comma ‘dtGT, attenuating GT...’

Answer: This was changed accordingly.

Line 138: ‘Not only is MpkA involved in...’

Answer: This was changed accordingly.

Line 139: revise for clarity

Answer: The sentence was changed to (Page 5, lines 136 to 140): “Not only is MpkA involved in GT self-protection but also other protein kinases are involved in this process since a screening of a collection of 110 non-essential protein kinase deletion mutants revealed six and one mutants as more GT-susceptible and –resistant, respectively”.

Line 154: watch consistency for labeling hours (e.g., hr, hrs, h, hs, hours). Adapt throughout. Also use consistent labeling for ‘minutes’ or ‘min’

Answer: Hour(s) is now shown as “h” while Minute(s) is now shown as “min”.

Line 185: end parentheses needed

Answer: This was changed accordingly.

Line 203: ‘introduce’

Answer: This was changed accordingly.

Line 218: missing closed square bracket

Answer: This was changed accordingly.

Line 267: suggest rephrasing '133 proteins that interact with...' as identification from immunoprecipitation does not imply direct interactions but indirect or complexed proteins may be identified. 'Associate with' may be more appropriate until interactions are directly confirmed.

Answer: This was changed accordingly.

Line 265: what does 'subtraction' refer to? Were background intensity values subtracted from protein intensity values or were background protein IDs removed from further analysis?

Answer: Background proteins unspecifically immunoprecipitated from the wild-type were removed from the analysis. This is explained now in the text (Page 7, lines 215 to 218): "During 24 to 72 h GT production, after subtraction of the proteins non-specifically immunoprecipitated of the wild-type strain, 133 proteins that associate with GliT:GFP and 203 proteins with GtmA:GFP were identified (Figure 5A)."

Line 272: were background proteins used for significance calculations (i.e., abundance changes considered for background proteins or presence/absence)?

Answer: Background proteins, i.e., proteins that were non-specifically immunoprecipitated of the wild-type strain, were removed from the analysis and not considered for significance calculation.

Line 283: remove duplicate 'including'

Answer: This was changed accordingly.

Line 340: proteins will be produced; genes will show changes in expression. Suggest revising use of 'expression' to describe production of the protein.

Answer: We thank the reviewer for the observation. Accordingly, we changed the sentence to (Page 10, lines 292 to 296): "The GliT:GFP and GtmA:GFP proteins are highly produced from 24 to 72 h while both proteins have increased production when the strains are exposed to GT (5 µg/mL) for 3 h; however, the GliT production is much higher than GtmA:GFP (Figure 6D)."

Line 357: does MpkA phosphorylate

Answer: Please, see answer below.

Line 337: what is meant by 'MpkA phosphorylation is increased...'? Frequency of phosphorylation or changes in abundance of phosphorylated proteins? If the latter, are phosphorylation-induced abundance changes normalized against changes in the total proteome between the tested conditions? Also relevant for line 387.

Answer: We thank the reviewer for the comments. We are actually talking about the phosphorylated forms of MpkA, as assessed by a western blot with commercial antibodies that can recognize phosphorylated and non-phosphorylated forms of MpkA. To avoid misunderstandings, the sentence was changed to (Page 10, lines 291 to 292): “The MpkA is more phosphorylated during GT production and self-protection (Figure 6D)”.

Line 392: check units for GT

Answer: This was changed accordingly.

Line 628: samples transferred to 2 mL HPLC vials (line 617) but 5 mL was injected into the mass spectrometer. Please clarify.

Answer: We thank the reviewer for the comment. This is wrong and the sentence was changed to (Page 21, line 686): “The injection volume was 5 μ L”.

Line 714: how many mice?

Answer: The number of mice were added to the section (Page 24, lines 784 to 786): Mice (10 per treatment) were immunosuppressed with cyclophosphamide (150 mg per kg of body weight), which was administered intraperitoneally on days -4, -1, and 2, prior to and post infection”.

Line 768: define Stage Tips (STAGE tips) and add reference

Answer: We have defined what StageTips are (Page 26, lines 826 to 832): “(StageTips were used; they are centrifugeable, compatible with automates for high throughput screening, and useful for biomolecular applications of peptide desalting and proteomics ,ZipTip, EMD Millipore, catalog number ZTC18S960)”.

Methods: a P-value for significance is reported in the Results but no reference to data analysis, bioinformatics, or statistical testing is included in the Methods. This needs to be revised.

Answer: We thank the reviewer for the suggestion. We have added a section for statistical analysis in the Methods (Page 26, lines 849 to 853): “**Statistical analysis.** Grouped column plots with standard deviation error bars were used for representations of data. For comparisons with data from wild-type or control conditions, we performed one-tailed, paired *t* tests or one-way analysis of variance (ANOVA). All statistical analyses and graphics building were performed by using GraphPad Prism 5.00 (GraphPad Software)”.

Line 1140: YUU is mentioned but not included elsewhere in the manuscript. Needs to be revised.

Answer: We have added the YUU composition to the Methods section of the manuscript (Page 19, lines 590 to 595): “Strains were grown at 37 °C. Conidia of *A. fumigatus* and *A. nidulans* were grown on complete medium (YG) [2% (w/v) glucose, 0.5% (w/v) yeast extract, trace elements] or minimal media (MM) [1% (w/v) glucose, nitrate salts, trace elements, pH 6.5]. Solid YG and MM were the

same as described above with the addition of 2% (w/v) agar. When necessary, uridine and uracil (1.2 g/L) were added (YAG+UU=YUU or MM+UU=MMUU)”.

Table 1: revise use of red and green given challenges in distinguishing colors for readers.

Answer: We thank the reviewer for the suggestion. We have changed the colours for blue and yellow in Table 1 (now Table 2).

Table 2: suggest revising title unless this is based on fulling confirmed interacting proteins

Answer: We thank the reviewer for the suggestion. The title of the Table 2 (now Table 1) was changed to (Page 41): “Table 1 – Proteins that are possibly associated with *A. fumigatus* GliT:GFP and GtmA:GFP under GT production.”

Figures: for microscopy images, how many images were taken and used for the analysis?

Answer: We thank the reviewer for this comment. In the Figures 1 and 2, we have counted two repetitions of 45 germlings. This is now shown at the legend of the Figures 1a and 2a (Page 37, lines 1179 to 1188): “**Figure 1 – GliT:GFP and GtmA:GFP have enriched vacuolar localization during GT production.** GliT:GFP and GtmA:GFP germlings were grown in liquid Czapek-dox medium for 24 h at 37 °C. Cell tracker Blue CMAC (CellTracker Blue CMAC Dye (7-amino-4-chloromethylcoumarin) was used for vacuolar staining. (a) The number of GliT:GFP and GtmA:GFP germlings that co-localized with Cell tracker Blue CMAC (CellTracker Blue CMAC Dye (7-amino-4-chloromethylcoumarin) that was used for vacuolar staining were determined. We have counted two independent experiments with 45 germlings for each strain and the results were expressed as the % of GFP that co-localizes with CMAC. Representative images are shown in (b) and (c). Bars, 5 µm”.

(Pages 37, lines 1190 to 1201): “**Figure 2 - GliT:GFP and GtmA:GFP localize in the cytoplasm, endoplasmic reticulum, and small vesicle structures during GT self-protection.** GliT:GFP and GtmA:GFP germlings were grown in liquid MM for 24 h at 37 °C and exposed to GT 3 µg/ml for 2 h and co-stained with either CMAC, ER-Tracker Blue-White DPX, or FM4-64 (a) The number of GliT:GFP and GtmA:GFP germlings that co-localized with CMAC, ER-tracker, or FM4-64 staining were determined. We have counted two independent experiments with 45 germlings for each strain and the results were expressed as the average (%) of GFP germlings ± standard deviation that co-localizes with vacuoles (CMAC), Endoplasmic Reticulum, ER (ER-tracker) or vesicles (FM4-64). The percentage of germlings with punctuated structures of GliT:GFP and GtmA:GFP were also determined. Representative images are shown in (b), (c), (d), (e), (f), and (g). Bars, 5 µm.”

And in the text (Pages 6 and 7, lines 165 to 184 and 185 to 191, respectively): “After 24 h in GT-production conditions, the wild-type has not shown any auto-

fluorescence in the GFP excitation and emission spectra (**Figures 1a and 1b**) while the GliT:GFP and GtmA:GFP were localized in the cytoplasm (100 % of the germlings) and had enriched localization in 89.44 and 96.57 % of the germlings, respectively (**Figure 1a**) of structures that resemble to vacuoles (**Figures 1b and 1c**). This was confirmed by co-staining with CellTracker Blue CMAC Dye (7-amino-4-chloromethylcoumarin; <https://www.thermofisher.com/order/catalog/product/C2110>) which accumulates in the vacuoles (**Figures 1b and 1c**). GliT:GFP and GtmA:GFP germlings were grown for 24 h in liquid MM and exposed to GT 3 µg/ml for 2 h and co-stained with either CMAC, ER-Tracker, or FM4-64 Dye [*N*-(3-Triethylammoniumpropyl)-4-(6-(4-(Diethylamino) Phenyl) Hexatrienyl) Pyridinium Dibromide] (**Figure 2**). ER-Tracker Blue-White DPX dye is a photostable probe that is selective for the endoplasmic reticulum (ER) in live cells (<https://www.thermofisher.com/order/catalog/product/E12353>). FM4-64 is recommended for staining vacuolar membranes and for studying the endocytic pathway ³⁷; <https://www.thermofisher.com/order/catalog/product/T13320>). Upon GT exposure, GliT:GFP accumulates in the cytoplasm, but not in the GT-free control, without any co-localization with CMAC (**Figures 2a and 2b**) but with 100 % germlings with ER-tracker (**Figures 2a and 2c**) and 100 % germlings with FM4-64 (**Figure 2a and 2d**). GtmA:GFP showed no fluorescence signal in the control but when the germlings were exposed to GT, there was diffuse and 100 % germlings with punctuated distribution in the cytoplasm (**Figures 2a, 2e, 2f, and 2g**). These punctuated structures were not observed in the GliT:GFP (**Figure 2a**). Upon GT exposure, GtmA:GFP did not co-localize with CMAC (**Figures 2a and 2e**), but 5 % germlings co-localized with ER-tracker (**Figures 2a and 2f**) and 100 % germlings with FM4-64 (**Figure 2a and 2g**).

Figures: for mass spectrometry information, if statistical testing for p-value determination was used, why are the analyses all qualitative and based on Venn diagrams and protein ID counts?

Answer: For mass spectrometry analysis we have not performed *p* tests. We used three biological replicates for each GliT:GFP and GtmA:GFP for each time points (three for 24 hours samples, three for 48 and three for 72 hours samples). We used untagged protein pulldowns to filter out unspecific proteins. What has been left after filtering was presented in final tables. Final tables and final protein numbers for each purification represents all specific proteins found in three purifications. We did not use *p* tests because we look at presence or absence of the proteins. This analysis has now been explained and cited within the methods section.

(Pages 25 and 26, lines 801 to 821 and 822 to 832, respectively): "**GFP or HA-Tag Protein Purification and Identification by LC-MS/MS**. To precipitate GFP/HA-tag labeled strains, protein crude extracts were prepared from cultures grown for 24, 48 and 72 h in Czapek Dox or in MM followed by gliotoxin treatment (5 µg / mL). Crude protein extracts from mycelia were obtained by extraction from ground mycelia with B250 buffer (250 mM NaCl, 100 mM Tris-HCl pH 7.5, 10% glycerol, 1 mM EDTA and 0.1% NP-40) supplemented with 1.5 mL/L 1 M DTT, 2

tablets/100 mL complete-mini protease inhibitor cocktail EDTA-free (Roche), 3 mL/L 0.5 M Benzamidine, 10 mL/L phosphatase inhibitors 100× (10 M NaF, 5 M Na Vanadate, 8 M β- glycerol phosphate), and 10 mL/L 100 mM PMSF. Extracts were centrifuged at 13,000 g for 20 min at 4°C, and the supernatant was collected into a new eppendorf. The same amount of protein for each sample was added to 20 µl of Magnetics GFP-trap beads or to Dynabeads Protein A (Thermo Fisher Scientific) previously incubated with monoclonal anti-HA antibody (Sigma). Cell extracts and beads were incubated with shaking at 4 °C for 4 h. After incubation, the magnetics beads were collected using magnetic hack and washed according to manufacturer's instructions. To release the proteins from the beads, samples were incubated with sample buffer and boiled at 98 °C for 5 min. Western Blot assays was carried out as described above. For the LC-MS/MS identification, the washed beads were re-suspended in 50 mM ammonium bicarbonate solution. Proteins were reduced with 10 mM DTT (DL-Dithiothreitol–Sigma-Aldrich) for 20 min at 56°C, alkylated with 40 mM iodoacetamide (Sigma-Aldrich) for 15 min at room temperature in the dark, and digested with trypsin (Promega) in the ratio 1:50 (µg trypsin/µg protein) at 37°C overnight. The digestion was stopped by addition of trifluoroacetic acid (TFA) to reach a final concentration of 1% and then the sample was desalted with C18 columns (StageTips were used; they are centrifugeable, compatible with automates for high throughput screening, and useful for biomolecular applications of peptide desalting and proteomics). Interaction partners of GliT and GtmA were analysed in at least three biological replicates filtered for unspecific peptides identified with AFUXX (untagged control strain). Peptides were run in LC-MS/MS and analysed as described in detail in ⁷⁵.

Reviewer #3 (Remarks to the Author):

This is an interesting work exploring the intracellular detoxification pathways that protect an important human fungal pathogen, *Aspergillus fumigatus*, from toxicity induced during production of the model mycotoxin gliotoxin. The authors generated transgenic strains to study intracellular trafficking and distribution of two self-protective GT detoxification enzymes encoding for an oxidoreductase and a methyltransferase, GliT and GtmA respectively. They performed a significant amount of work using state of art molecular biology and biochemical tools to identify an important role of peroxisomes in GT detoxification, and an important role of MAP kinase signaling in GT production and GliT and GtmA trafficking. Overall, the work lacks sufficient mechanistic insight on the molecular interaction of GT with the detoxification enzymes of interest and the cellular organelles that are involved in this process.

Answer: We thank very much the reviewer for the comments and suggestions. They have considerably improved the quality of our manuscript.

In particular, a direct role of peroxisomes in intracellular trafficking and activity of GliT and GtmA is not shown (the transgenic molecules fail to colocalize with peroxisomes)

The protective activity of peroxisomes against GT is not explored in detail. The mechanistic link between peroxisomes and MAP kinases is not shown. The precise mechanism of intracellular trafficking of the two self-protective enzymes is not shown and the microscopy methods that are utilized lack sufficient precision to capture these processes (e.g., need for confocal and super-resolution microscopy approaches)

Answer: We thank the reviewer for the comments and suggestions. The main goal of our work was to understand how *A. fumigatus* GliT and GtmA are regulated and investigate their sub-cellular localization upon gliotoxin production and self-protection. Our results clearly show that the *A. fumigatus* mitogen-activated protein (MAP) kinase MpkA is important for *gliT* and *gtmA* mRNA accumulation and protein production. MpkA is essential for gliotoxin production and self-defense. Peroxisomes are not involved in the intracellular trafficking of GliT:GFP and GtmA:GFP upon gliotoxin production and self-defense. Although the main goal of our work was to understand GliT and GtmA regulation and sub-cellular localization upon gliotoxin production and self-defense, we also provided as an additional contribution the discovery that peroxisomes play a new role in *A. fumigatus* gliotoxin self-protection and production. To highlight these two important independent discoveries, we reorganized the sub-section “***Aspergillus spp* peroxisome receptors PexE^{Pex5} and PexG^{Pex7} are important for GT self-protection**”, placing it as the last sub-section in the **Results** section. This was also highlighted in the Introduction section (Pages 5 and 6, lines 146 to 151 and 152 to 153, respectively): “In addition to GliT and GtmA regulation and sub-cellular localization, we also investigated the involvement of peroxisomes on GT

self-protection, we characterized the PexE^{Pex5} and PexG^{Pex7} peroxisome receptors, and how they mediate GT self-protection. We demonstrate that both receptors are involved in the regulation of GT production but only *A. fumigatus* $\Delta pexE$ mutant is GT-sensitive and has attenuated virulence in a murine model of invasive pulmonary aspergillosis (IPA).”

We discovered that peroxisomes play a new role in *A. fumigatus* gliotoxin self-protection and production. We clearly showed by using genetic means that *A. fumigatus* *pex5* and *pex7* receptor genes played important roles in gliotoxin production and self-protection, as discussed, stated and speculated in the Discussion section (Page 18, lines 556 to 577): “We investigated the influence of peroxisomes on GT production and self-protection by characterizing *A. fumigatus* *pexE* and *pexG* deletion mutants. Interestingly, while $\Delta pexE$ produces very low levels of GT and bmGT and is very sensitive to GT, the $\Delta pexG$ mutant is as sensitive to GT as the wild-type and has increased production of GT. These results suggest that both receptors play a role in GT production and self-protection. Although there is increased production of *gli* genes and *gtmA* mRNA levels in the $\Delta pexG$ mutant in both non-inducing and inducing conditions, the increased GT production in this mutant only occurs in inducing conditions. These results suggest that PexG modulates the production of GT but this modulation only happens through the same signal transduction pathways that coordinate the GT induction. It remains to be investigated which proteins are transported to the peroxisomes via PexG and are involved in the modulation of GT production under GT inducing conditions. What could be the peroxisomal mechanisms influencing GT production and self-protection? We detected increased sensitivity of $\Delta pexG$ to menadione that could suggest peroxisomes are required for the optimal oxireduction cellular environment necessary for GT production and self-protection. We were not able to assign any other phenotype that could impact GT self-protection in these mutants, such as sulfur source utilization. Interestingly, PexE and PexG are also important for self-defense in *A. nidulans*, a non GT producer. Our results indicate an important role for peroxisomes in GT production and self-defense, and virulence”.

We explained in more details our motivation to investigate the role played by peroxisomes in this process and performed several additional experiments measuring if the growth in gliotoxin-inducing or not-inducing conditions could increase the number of peroxisomes in the germlings.

(Page 12, lines 355 to 364): “Peroxisomes are responsible for detoxification, catabolism of linear and branched-chain fatty acid, and removal of H₂O₂ by catalases³⁹⁻⁴². Many SMs are initially synthesized in the peroxisomes, such as aflatoxin and sterigmatocystin, as biosynthetic steps of many other SM pathways, such as paxilline, AK-toxin, penicillin, cephalosporin and some siderophores^{16, 43}. The accumulation of GT and dtGT has a great impact on the generation of reactive oxygen species by the cycling between the oxidised (disulfide) and reduced (dithiol) forms of the molecule³. We speculate that due to the important role played by peroxisomes in the oxiredox equilibrium of the cell, they could also be involved in *A. fumigatus* GT production and self-protection”.

We performed extensive quantification of all the microscopies (including the relevant techniques, controls, quantification and statistical analyses where appropriate).

The term “organelle” that is frequently used to describe localization of the enzymes is not appropriate. The authors even fail to use specialized markers to describe ER. Speculation for a non-canonical pathway of secretion of the enzymes is mentioned in the text. However, this has not been experimentally shown.

Answer: We thank very much the reviewer for the comments. The term “organelle” was removed from the manuscript.

We thank the reviewer for the suggestion of using specialized markers to describe ER. We have now used the ER-Tracker Blue-White DPX, a photostable probe that is selective for the endoplasmic reticulum (ER) in live cells (<https://www.thermofisher.com/order/catalog/product/E12353>). We have performed additional microscopic experiments and the text was changed to (Pages 6 and 7, lines 165 to 184 and 185 to 191, respectively): “After 24 h in GT-production conditions, the wild-type has not shown any auto-fluorescence in the GFP excitation and emission spectra (**Figures 1a and 1b**) while the GliT:GFP and GtmA:GFP were localized in the cytoplasm (100 % of the germlings) and had enriched localization in 89.44 and 96.57 % of the germlings, respectively (**Figure 1a**) of structures that resemble to vacuoles (**Figures 1b and 1c**). This was confirmed by co-staining with CellTracker Blue CMAC Dye (7-amino-4-chloromethylcoumarin; <https://www.thermofisher.com/order/catalog/product/C2110>) which accumulates in the vacuoles (**Figures 1b and 1c**). GliT:GFP and GtmA:GFP germlings were grown for 24 h in MM and exposed to GT 3 µg/ml for 2 h and co-stained with either CMAC, ER-Tracker, or FM4-64 Dye [*N*-(3-Triethylammoniumpropyl)-4-(6-(4-(Diethylamino) Phenyl) Hexatrienyl) Pyridinium Dibromide] (**Figure 2**). ER-Tracker Blue-White DPX dye is a photostable probe that is selective for the endoplasmic reticulum (ER) in live cells (<https://www.thermofisher.com/order/catalog/product/E12353>). FM4-64 is recommended for staining vacuolar membranes and for studying the endocytic pathway ³⁷; <https://www.thermofisher.com/order/catalog/product/T13320>). Upon GT exposure, GliT:GFP accumulates in the cytoplasm, but not in the GT-free control, without any co-localization with CMAC (**Figures 2a and 2b**) but with 100 % germlings with ER-tracker (**Figures 2a and 2c**) and 100 % germlings with FM4-64 (**Figure 2a and 2d**). GtmA:GFP showed no fluorescence signal in the control but when the germlings were exposed to GT, there was diffuse and 100 % germlings with punctuated distribution in the cytoplasm (**Figures 2a, 2e, 2f, and 2g**). These punctuated structures were not observed in the GliT:GFP (**Figure 2a**). Upon GT exposure, GtmA:GFP did not co-localize with CMAC (**Figures 2a and 2e**), but 5 % germlings co-localized with ER-tracker (**Figures 2a and 2f**) and 100 % germlings with FM4-64 (**Figure 2a and 2g**).”

What we meant by “non-canonical pathway of secretion” is the fact that GT is not secreted exclusively by the transporter GliA. We are not interested in understanding *A. fumigatus* non-canonical pathways of secretion of GT or any other secondary metabolites. We are interested in *A. fumigatus* GT self-

protection and how this affects GT production. To avoid mis-interpretations, we removed any mention to “non-canonical pathways” and emphasized secretion by vesicles. Accordingly, the following sentences were changed in the manuscript

(Pages 5, lines 127 to 128): “We observe that GT is not only secreted by GliA but also through a vesicle-driven secretion mechanism”.

(Page 16, lines 488 to 490): “Since the fungus is exposed to a high GT concentration, part of the free GT is immediately transported to these vesicles, metabolized or not to dtGT and bmGT, and secreted through vesicles”.

(Page 15, lines 472 to 473): “(ii) about 12 and 3 % of the total GT and bmGT, respectively, are secreted as vesicles”

(Page 18, lines 578 to 580): “In summary, our work demonstrates another mechanism of GT self-protection through a possible GT-dtGT-bmGT storage in the vacuolar system while part of GT-bmGT is further directly secreted into vesicles”.

Reviewer #4 (Remarks to the Author):

This study investigates the controlled production of mycotoxin gliotoxin, a toxin produced by the pathogenic filamentous fungus, *Aspergillus fumigatus*. Mycotoxin gliotoxin (MG) is important for virulence, making this an important pathway to understand. The authors investigate multiple aspects of the cell biology of MG production, which is an important research direction for the field. However, in general the imaging experiments are not quantified, something that is unacceptable for publication as only representative images are shown. Controls are missing in multiple experiments and parts of the paper are confusing. Overall, this read like many different stories thrown into a single manuscript with not enough rigor in many places. This is a very interesting topic, but unfortunately, the data shown do not convince me of the authors conclusions.

Answer: We thank very much the reviewer for the comments and suggestions. We follow all your suggestions performing additional experiments and adding quantification of the imaging experiments. We believe these comments and suggestions have dramatically improved the quality of the manuscript.

Specific comments:

- Figures 1 and 2 present localization experiments, but no quantification is performed. This is not acceptable for publication.

Answer: We thank the reviewer for this comment. In the Figures 1 and 2, we have counted two repetitions of 45 germlings. This is now shown at the legend of the Figures 1a and 2a (Page 18, lines 1179 to 1188): **“Figure 1 – GliT:GFP and GtmA:GFP have enriched vacuolar localization during GT production.** GliT:GFP and GtmA:GFP germlings were grown in liquid Czapek-dox medium for 24 h at 37 °C. Cell tracker Blue CMAC (CellTracker Blue CMAC Dye (7-amino-4-chloromethylcoumarin) was used for vacuolar staining. (a) The number of GliT:GFP and GtmA:GFP germlings that co-localized with Cell tracker Blue CMAC (CellTracker Blue CMAC Dye (7-amino-4-chloromethylcoumarin) that was used for vacuolar staining were determined. We have counted two independent experiments with 45 germlings for each strain and the results were expressed as the % of GFP that co-localizes with CMAC. Representative images are shown in (b) and (c). Bars, 5 μm”.

(Page 18, lines 1190 to 1201): **“Figure 2 - GliT:GFP and GtmA:GFP localize in the cytoplasm, endoplasmic reticulum, and small vesicle structures during GT self-protection.** GliT:GFP and GtmA:GFP germlings were grown in liquid MM for 24 h at 37 °C and exposed to GT 3 μg/ml for 2 h and co-stained with either CMAC, ER-Tracker Blue-White DPX, or FM4-64 (a) The number of GliT:GFP and GtmA:GFP germlings that co-localized with CMAC, ER-tracker, or FM4-64 staining were determined. We have counted two independent experiments with 45 germlings for each strain and the results were expressed as the average (%) of GFP germlings ± standard deviation that co-localizes with vacuoles (CMAC), Endoplasmic Reticulum, ER (ER-tracker) or vesicles (FM4-64). The percentage of germlings with punctuated structures of GliT:GFP and GtmA:GFP were also

determined. Representative images are shown in (b), (c), (d), (e), (f), and (g). Bars, 5 μ m.”

And in the text (Pages 6 and 7, lines 165 to 184 and 185 to 191, respectively): “After 24 h in GT-production conditions, the wild-type has not shown any auto-fluorescence in the GFP excitation and emission spectra (**Figures 1a and 1b**) while the GliT:GFP and GtmA:GFP were localized in the cytoplasm (100 % of the germlings) and had enriched localization in 89.44 and 96.57 % of the germlings, respectively (**Figure 1a**) of structures that resemble to vacuoles (**Figures 1b and 1c**). This was confirmed by co-staining with CellTracker Blue CMAC Dye (7-amino-4-chloromethylcoumarin; <https://www.thermofisher.com/order/catalog/product/C2110>) which accumulates in the vacuoles (**Figures 1b and 1c**). GliT:GFP and GtmA:GFP germlings were grown for 24 h in liquid MM and exposed to GT 3 μ g/ml for 2 h and co-stained with either CMAC, ER-Tracker, or FM4-64 Dye [*N*-(3-Triethylammoniumpropyl)-4-(6-(4-(Diethylamino) Phenyl) Hexatrienyl) Pyridinium Dibromide] (**Figure 2**). ER-Tracker Blue-White DPX dye is a photostable probe that is selective for the endoplasmic reticulum (ER) in live cells (<https://www.thermofisher.com/order/catalog/product/E12353>). FM4-64 is recommended for staining vacuolar membranes and for studying the endocytic pathway ³⁷; <https://www.thermofisher.com/order/catalog/product/T13320>). Upon GT exposure, GliT:GFP accumulates in the cytoplasm, but not in the GT-free control, without any co-localization with CMAC (**Figures 2a and 2b**) but with 100 % germlings with ER-tracker (**Figures 2a and 2c**) and 100 % germlings with FM4-64 (**Figure 2a and 2d**). GtmA:GFP showed no fluorescence signal in the control but when the germlings were exposed to GT, there was diffuse and 100 % germlings with punctuated distribution in the cytoplasm (**Figures 2a, 2e, 2f, and 2g**). These punctuated structures were not observed in the GliT:GFP (**Figure 2a**). Upon GT exposure, GtmA:GFP did not co-localize with CMAC (**Figures 2a and 2e**), but 5 % germlings co-localized with ER-tracker (**Figures 2a and 2f**) and 100 % germlings with FM4-64 (**Figure 2a and 2g**).”

- The rationale for turning to peroxisomes is not made clear by the authors and the authors do not provide evidence that links GliT or GtmA to peroxisomes, yet state that in their title. This is an interesting line of research but needs to be addressed with quantification for publication.

Answer: We thank the reviewer for the comments and suggestions. The main goal of our work was to understand how *A. fumigatus* GliT and GtmA are regulated and investigate their sub-cellular localization upon gliotoxin production and self-protection. Our results clearly show that the *A. fumigatus* mitogen-activated protein (MAP) kinase MpkA is important for *gliT* and *gtmA* mRNA accumulation and protein production. MpkA is essential for gliotoxin production and self-defense. Peroxisomes are not involved in the intracellular trafficking of GliT:GFP and GtmA:GFP upon gliotoxin production. Although the main goal of our work was to understand GliT and GtmA regulation and sub-cellular localization upon gliotoxin production and self-defense, we also provided as an additional contribution the discovery that peroxisomes play a new role in *A. fumigatus* gliotoxin self-protection and production. To highlight these two important independent discoveries, we reorganized the sub-section “***Aspergillus***”

spp peroxisome receptors PexE^{Pex5} and PexG^{Pex7} are important for GT self-protection”, placing it as the last sub-section in the **Results** section. This was also highlighted in the Introduction section (Pages 5 and 6, lines 146 to 151 and 152 to 153, respectively): “In addition to GliT and GtmA regulation and sub-cellular localization, we also investigated the involvement of peroxisomes on GT self-protection, we characterized the PexE^{Pex5} and PexG^{Pex7} peroxisome receptors, and how they mediate GT self-protection. We demonstrate that both receptors are involved in the regulation of GT production but only *A. fumigatus* $\Delta pexE$ mutant is GT-sensitive and has attenuated virulence in a murine model of invasive pulmonary aspergillosis (IPA).”

We added now a rationale why it is important to investigate peroxisomes in the gliotoxin production and self-protection (Page 12, lines 355 to 364): “Peroxisomes are responsible for detoxification, catabolism of linear and branched-chain fatty acid, and removal of H₂O₂ by catalases³⁹⁻⁴². Many SMs are initially synthesized in the peroxisomes, such as aflatoxin and sterigmatocystin, as biosynthetic steps of many other SM pathways, such as paxilline, AK-toxin, penicillin, cephalosporin and some siderophores^{16, 43}. The accumulation of GT and dtGT has a great impact on the generation of reactive oxygen species by the cycling between the oxidised (disulfide) and reduced (dithiol) forms of the molecule³. We speculate that due to the important role played by peroxisomes in the oxiredox equilibrium of the cell, they could also be involved in *A. fumigatus* GT production and self-protection”.

The title of the manuscript refers to the involvement of vacuoles and peroxisomes in *Aspergillus fumigatus* gliotoxin production and self-protection. Our results show that vacuoles are involved (but not peroxisomes) in the intracellular trafficking of GliT:GFP and GtmA:GFP upon gliotoxin production. Actually, our work provided as a contribution the discovery that peroxisomes play a new role in *A. fumigatus* gliotoxin production that is not related to traffic and/or accumulation of GliT and GtmA upon gliotoxin production. This was discussed and explained in the discussion section. We clearly showed by using genetic means that *A. fumigatus pex5* and *pex7* receptor genes played important roles in gliotoxin production and self-protection, as discussed, stated and speculated in the Discussion section (Page 18, lines 556 to 577): “We investigated the influence of peroxisomes on GT production and self-protection by characterizing *A. fumigatus pexE* and *pexG* deletion mutants. Interestingly, while $\Delta pexE$ produces very low levels of GT and bmGT and is very sensitive to GT, the $\Delta pexG$ mutant is as sensitive to GT as the wild-type and has increased production of GT. These results suggest that both receptors play a role in GT production and self-protection. Although there is increased production of *gli* genes and *gtmA* mRNA levels in the $\Delta pexG$ mutant in both non-inducing and inducing conditions, the increased GT production in this mutant only occurs in inducing conditions. These results suggest that PexG modulates the production of GT but this modulation only happens through the same signal transduction pathways that coordinate the GT induction. It remains to be investigated which proteins are transported to the peroxisomes via PexG and are involved in the modulation of GT production under GT inducing conditions. What could be the peroxisomal mechanisms influencing GT production and self-protection? We detected increased sensitivity of $\Delta pexG$ to menadione that could suggest

peroxisomes are required for the optimal oxireduction cellular environment necessary for GT production and self-protection. We were not able to assign any other phenotype that could impact GT self-protection in these mutants, such as sulfur source utilization. Interestingly, PexE and PexG are also important for self-defense in *A. nidulans*, a non GT producer. Our results indicate an important role for peroxisomes in GT production and self-defense, and virulence”.

- Throughout the localization experiments, important controls are missing. For example, is vacuole autofluorescence an issue?

Answer: We thank the reviewer for the comment. The wild-type control without any GFP fusion was added to the Figure 1. We mentioned in the text (Page 6, lines 165 to 170): “After 24 h in GT-production conditions, the wild-type has not shown any auto-fluorescence in the GFP excitation and emission spectra (**Figures 1a and 1b**) while the GliT:GFP and GtmA:GFP were localized in the cytoplasm (100 % of the germlings) and had enriched localization in 89.44 and 96.57 % of the germlings, respectively (**Figure 1a**) of structures that resemble to vacuoles (**Figures 1b and 1c**).”

- The number of hyphae examined is not listed and again quantification is not performed.

Answer: Please, see the answer above to this question. We now quantified all the microscopic experiments

- In Figure 2 the co-localization with FM4-64 and GliT is hard to see. Are the GtmA-GFP puncta dynamic? The hypha in Fig 2F second row (+GT) looks damaged compared to the control and the right hypha.

Answer: We thank the reviewer for the questions. We agree with the reviewer that co-localization with FM4-64 and GliT:GFP is hard to see. The same problem happens with the co-localization with the ER-tracker. This is due to the fact that GliT:GFP has a more diffuse distribution. However, some spots are clearly co-localizing with FM4-64 and ER-tracker by looking carefully at the figures.

Unfortunately, we do not know if the GtmA:GFP puncta are dynamic. However, they were not observed in the GliT:GFP strain but only in GtmA:GFP strain.

The hypha in Figure 2F is not damaged. There are two germlings together overlapping by the round initial structures of the germling. However, it was removed from the manuscript because the Hoechst was replaced by the ER-tracker.

- In Figure 3 the authors identify GliF and GliN in purified vesicles. Is it possible to localize both proteins with GFP in cells under GT production and self-protection conditions?

Answer: We thank the reviewer for the suggestion. That is a very interesting experiment. However, if the reviewer does not mind, we would prefer not to

perform such experiment in this manuscript considering the large investment and time in the construction of these strains.

- L215 is confusing since the authors only mention in the discussion that GtmA:GFP doesn't co-localize with mRFP:PTS1. It would be useful to include this result in the results section.

Answer: We apologize about the confusion. Actually what we meant by the sentence "We were not able to demonstrate the colocalization of GliT:GFP and GtmA:GFP with peroxisomal targeted RFP (mRFP:PTS1)." is that we have not performed the experiment of co-localization of mRFP:PTS1 and GliT:GFP and GtmA:GFP. In order to avoid misinterpretations, we removed this sentence from the manuscript.

- In Figure 4 the authors used a concentration of 30µg/ml GT for the inhibitor study compared to 3µg/ml for the microscopic localization of GliT & GtmA (Figure 2.). Is there a specific reason why? In addition, the *A. nidulans* samples were treated with 5µg/ml. Why the change in concentration?

Answer: We thank the reviewer for the comment. We have added a sentence explaining the reasons why different concentrations were used along the manuscript (Page 13, lines 398 to 401): "Notice that we are using different GT concentrations for microscopy (3 µg/ml at Figure 1) and *A. fumigatus* and *A. nidulans* radial growth experiments (30 µg/ml versus 5 µg/ml, Figure 4) because germlings are much more sensitive to GT than the mycelia and *A. nidulans* is more GT-sensitive than *A. fumigatus*."

- Can the authors comment on why the *A. fumigatus* pexE mutant is so sensitive to GT compared to the *A. nidulans* mutant?

Answer: We thank very much the reviewer for the comment. We have added a sentence commenting it (Page 14, lines 445 to 447): "Our results also indicate *A. fumigatus* and *A. nidulans* PexE homologs have different functions in the GT self-protection since *A. fumigatus* Δ pexE is much more sensitive to GT than *A. nidulans* Δ pexE strain".

- As the *A. fumigatus* pexG mutant shows a 25 to 50 fold increase in GT production - does it have better detoxification mechanisms? Would it make sense to localize GliT and GtmA in the mutant background? It might be interesting to see if the Δ pexG mutant also produces GT in non-inducing conditions.

Answer: We thank very much the reviewer for this interesting question. We decided to address it by two different experiments: (i) by measuring the GT production in non-inducing (MM) and inducing conditions (Czapek-dox medium) and (ii) by measuring the mRNA accumulation of all the *gli* and *gtmA* genes in non-inducing and inducing conditions. This is now described in the text (Page 14, lines 421 to 442) and in Supplementary Figure S4: "GT production is normally induced in Czapek-dox medium and *Aspergillus* minimal medium is regarded as a non-inducing condition³⁶. Due to the high levels of GT production by Δ pexG mutant, we comparatively investigated the mRNA levels of the *gli* genes and

gtmA, and the GT and bmGT production under inducing (grown in Czapek-dox medium) and non-inducing (grown in *Aspergillus* minimal medium) conditions (**Figure 7 and Supplementary Figure S4**). mRNA levels were expressed as the fold increase of each gene from the $\Delta pexG$ mutant divided by the wild-type: fold increase ratios equal to 1, below 1 or above 1 corresponded to similar, lower or higher mRNA levels in the $\Delta pexG$ mutant than in the wild-type (**Supplementary Figure S4**). In non-inducing conditions, all *gli* genes had higher than 1 expression ratios at 48 h production except for *gtmA* that had higher than 1 expression ratio at 72 h (**Supplementary Figure S4a**). In inducing conditions, all *gli* genes and *gtmA* had higher than 1 expression ratios at 24 h (**Supplementary Figure S4b**). In both conditions, most of the *gli* genes and *gtmA* had higher than 1 expression ratios in more than a single timepoint (**Supplementary Figure S4a and S4b**). However, both the wild-type and $\Delta pexG$ mutants did not produce GT under non-inducing conditions but as expected $\Delta pexG$ had much higher GT production than the wild-type under inducing conditions (**Supplementary Figure S4c**). Taken together, these results suggest that although PexG can affect *gli* genes and *gtmA* mRNA levels accumulation under both non-inducing and inducing conditions, the increased GT production in the $\Delta pexG$ only occurs under inducing conditions.”

And in the Discussion section (Page 18, lines 561 to 569): “Although there is increased production of *gli* genes and *gtmA* mRNA levels in the $\Delta pexG$ mutant in both non-inducing and inducing conditions, the increased GT production in this mutant only occurs in inducing conditions. These results suggest that PexG modulates the production of GT but this modulation only happens through the same signal transduction pathways that coordinate the GT induction. It remains to be investigated which proteins are transported to the peroxisomes via PexG and are involved in the modulation of GT production under GT inducing conditions.”

- Figure 6, in B the GT production was measured after 72 h. Given the strong growth defect of the *mpkA* mutant it might be necessary to extend the incubation time by multiple days.

Answer: We thank very much the reviewer for this excellent observation. Although the $\Delta mpkA$ mutant has a strong growth defect in solid medium with significantly reduced radial growth, surprisingly its growth in liquid Czapek-dox medium after 72 h is comparable to the wild-type. This observation was added to the manuscript (Page 9, lines 278 to 283): “Although the $\Delta mpkA$ mutant has a strong growth defect in solid medium with significantly reduced radial growth, surprisingly its growth in liquid Czapek-dox medium after 72 h is not statistically different from the wild-type (wild-type, 0.321 ± 0.014 mg and $\Delta mpkA$, 0.303 ± 0.005 mg). After 72 h growth in Czapek-dox medium the $\Delta mpkA$ mutant does not produce either GT or bmGT (Figure 6B).

Typos:

L171: CMCA -> CMAC

Answer: This was changed accordingly.

L175: Figures 2C and 2D -> Fig 2C-E?

Answer: These figures were changed to Supplementary Figures S1C to S1E.

L458: PST1 -> PTS1

Answer: This was changed accordingly.

REVIEWER COMMENTS

Reviewer #3 (Remarks to the Author):

The authors provide new experimental data that sufficiently address my previous comments and clarify most of the remaining issues. The manuscript has improved in terms of clarity and conclusions

Reviewer #4 (Remarks to the Author):

The quality of this manuscript has not significantly improved. It is difficult to read and tries to merge 3-4 different stories into one paper. The imaging data in Figures 1 and 2 is of poor quality and the methods used to quantify the images are poorly explained and were not repeated in triplicate, which would be required for statistical analysis. These figures should be removed from the paper. Figure 3 is also quite preliminary and requires additional follow up experiments. My advice would be to re-focus the paper on the identification and characterization of MpkA (Figures 4 and 5) and expand on these findings. The peroxisome data is intriguing, but over-stated and preliminary without clear explanations of how the experiments were conducted or how the data was analyzed. The title and abstract do not support the conclusions of the paper. Unfortunately, I cannot support publication of this work in its current form.

Answers to the Reviewer

Reviewer #4 (Remarks to the Author):

1) The quality of this manuscript has not significantly improved. It is difficult to read and tries to merge 3-4 different stories into one paper.

Answer: We apologize about it. We tried to follow all the reviewer's suggestions to improve it. We now decided to focus on the MpkA story as suggested by the reviewer, changing the title ("*Aspergillus fumigatus* mitogen-activated protein kinase MpkA is involved in gliotoxin production and self-protection"), and attenuating the conclusions about the role played by peroxisomes. As requested by the reviewer, we also removed the Figure 3 and all the associated data from the manuscript. We agree the data are too preliminary and requires additional follow up experiments. We hope the manuscript is now appropriate for the reviewer.

2) The imaging data in Figures 1 and 2 is of poor quality and the methods used to quantify the images are poorly explained and were not repeated in triplicate, which would be required for statistical analysis. These figures should be removed from the paper.

Answer: We thank the reviewer for the comments. We would prefer not to remove the figures 1 and 2 from the manuscript. If the reviewer does not mind, we would prefer to keep these Figures. The data from Figure 1 was now repeated three times. The methods used to quantify the images in the Figures 1 and 2 are now described in the Methods section (Page 19, lines 602 to 608), and in the corresponding legends of the Figures 1 and 2: "In the GT production assay, germlings were assigned as positively present in the vacuoles if they have concomitant localization of GliT:GFP or GtmA:GFP in the same vacuole stained by CMC in the same germling. In the GT self-protection assay, germlings were assigned as positively present in the vacuoles, endoplasmic reticulum, vesicles, or endomembranes if at least one of these structures was co-localizing with GliT:GFP or GtmA:GFP in the same germling."

3) Figure 3 is also quite preliminary and requires additional follow up experiments.

Answer: As suggested by the reviewer, Figure 3 and the associated data were removed from the manuscript.

4) My advice would be to re-focus the paper on the identification and characterization of MpkA (Figures 4 and 5) and expand on these findings.

Answer: We thank the reviewer for the suggestions. We have followed the reviewer's advices and re-focus on the identification and characterization of the MpkA. We decided to keep the protein kinase null library screening because this screening allowed us to identify SlnA as a receptor activating the MpkA

phosphorylation. We changed the title of the manuscript to: “*Aspergillus fumigatus* mitogen-activated protein kinase MpkA is involved in gliotoxin production and self-protection”

5) The peroxisome data is intriguing, but over-stated and preliminary without clear explanations of how the experiments were conducted or how the data was analyzed. The title and abstract do not support the conclusions of the paper. Unfortunately, I cannot support publication of this work in its current form.

Answer: We have attenuated the conclusions about the role played by peroxisomes during GT production and self-defense along the manuscript, including the abstract section. We also changed the title of the manuscript trying to focus on the MpkA story as suggested by the reviewer (“*Aspergillus fumigatus* mitogen-activated protein kinase MpkA is involved in gliotoxin production and self-protection”). It is not clear for us what the reviewer means by “without clear explanations of how the experiments were conducted or how the data was analyzed”. All the experiments are well-described in the results and methods section.